# Quantifying the Sensitivity of Inverse Reinforcement Learning to Misspecification

**Joar Skalse & Alessandro Abate**
Department of Computer Science
Oxford University
{joar.skalse,aabate}@cs.ox.ac.uk

## Abstract

Inverse reinforcement learning (IRL) aims to infer an agent's *preferences* (represented as a reward function $R$) from their *behaviour* (represented as a policy $\pi$). To do this, we need a *behavioural model* of how $\pi$ relates to $R$. In the current literature, the most common behavioural models are *optimality*, *Boltzmann-rationality*, and *causal entropy maximisation*. However, the true relationship between a human's preferences and their behaviour is much more complex than any of these behavioural models. This means that the behavioural models are *misspecified*, which raises the concern that they may lead to systematic errors if applied to real data. In this paper, we analyse how sensitive the IRL problem is to misspecification of the behavioural model. Specifically, we provide necessary and sufficient conditions that completely characterise how the observed data may differ from the assumed behavioural model without incurring an error above a given threshold. In addition to this, we also characterise the conditions under which a behavioural model is robust to small perturbations of the observed policy, and we analyse how robust many behavioural models are to misspecification of their parameter values (such as e.g. the discount rate). Our analysis suggests that the IRL problem is highly sensitive to misspecification, in the sense that very mild misspecification can lead to very large errors in the inferred reward function.

## 1 Introduction

Inverse reinforcement learning (IRL) is a subfield of machine learning that aims to develop techniques for inferring an agent's *preferences* based on their *actions* in a sequential decision-making problem (Ng & Russell, 2000). There are many motivations for IRL. One motivation is to use it as a tool for *imitation learning*, where the objective is to replicate the behaviour of an expert in some task (e.g. Hussein et al., 2017). In this context, it is not essential that the inferred preferences reflect the actual intentions of the expert, as long as they improve the imitation learning process. Another motivation for IRL is to use it as a tool for *preference elicitation*, where the objective is to understand an agent's goals or desires (e.g. Hadfield-Menell et al., 2016). In this context, it is of central importance that the inferred preferences reflect the actual preferences of the observed agent. In this paper, we are primarily concerned with this second motivation.

An IRL algorithm must make assumptions about how the preferences of the observed agent relate to its behaviour. Specifically, in IRL, preferences are typically modelled as a reward function $R$, and behaviour is typically modelled as a policy $\pi$. An IRL algorithm must therefore have a *behavioural model* that describes how $\pi$ is computed from $R$; by inverting this model, $R$ can then be deduced based on $\pi$. In the current literature, the most common behavioural models are *optimality*, *Boltzmann rationality*, or *causal entropy maximisation*. These behavioural models essentially assume that the observed agent behaves in a way that is (noisily) optimal according to its preferences.

One of the central difficulties in IRL is that the true relationship between a person's preferences and their actions in general is incredibly complex. This means that it typically is very difficult to specify a behavioural model that is perfectly accurate. For example, optimality, Boltzmann-rationality, and causal entropy maximisation are all very simple models that clearly do not capture all the nuances

of human behaviour.[1] This means that these behavioural models are *misspecified*, which raises the concern that they might systematically lead to flawed inferences if applied to real data.

In this paper, we study how robust IRL is to misspecification of the behavioural model. If an IRL algorithm that assumes a particular behavioural model is shown data from an agent whose behaviour violates the assumptions behind this model, then will the inferred reward function still be close to the true reward function? We will analyse this question mathematically, and provide several quantitative answers. In particular, we provide necessary and sufficient conditions that completely characterise what types of misspecification a wide class of behavioural models will tolerate. In addition to this, we also study two specific types of misspecification in more detail (namely, perturbation of the observed policy, and misspecification of the parameters in the behavioural model) and provide additional results that describe how they affect the quality of the inferred reward. Our analysis is highly general – it applies to the three behavioural models that are most common in the current IRL literature, but is also directly applicable to a much wider class of models.

The motivation behind this paper is to contribute towards a theoretically principled understanding of when IRL methods are applicable to the problem of preference elicitation. Human behaviour is very complex, and while we can create behavioural models that are more accurate, it will never be realistically possible to create a behavioural model that is totally free from misspecification. It is therefore crucial to have an understanding of how robust the IRL problem is to misspecification, and whether a small amount of misspecification leads to a proportionally small error in the inferred reward function. Our work aims to further our understanding of these question.

## 1.1 RELATED WORK

There are two previous papers that analyse how robust the IRL problem is to misspecified behavioural models; Hong et al. (2022) and Skalse & Abate (2023). Our work is more complete than these earlier works in several important respects. To start with, our problem setup is both more realistic, and more general. In particular, in order to quantify how robust IRL is to misspecification, we first need a way to formalise what it means for two reward functions to be "close". Skalse & Abate (2023) formalise this in terms of *equivalence relations*, under which two reward functions are either equivalent or not. As such, their analysis is somewhat blunt, and is unable to distinguish between small errors and large errors in the inferred reward. Hong et al. (2022) instead use the $\ell^2$-distance between the reward functions. However, this choice is also problematic, because two reward functions can be very dissimilar even though they have a small $\ell^2$-distance, and vice versa (cf. Section 2.2). By contrast, our analysis is carried out in terms of specially selected *metrics* on the space of all reward functions, which are backed by strong theoretical guarantees. Moreover, Hong et al. (2022) assume that there is a *unique* reward function that maximises fit to the training data, but this is violated in most real-world cases (Ng & Russell, 2000; Dvijotham & Todorov, 2010; Cao et al., 2021; Kim et al., 2021; Skalse et al., 2022; Schlaginhaufen & Kamgarpour, 2023). In addition to this, many of their results also assume "strong log-concavity", which is a rather opaque condition that is left mostly unexamined. Indeed, Hong et al. (2022) explicitly do not answer if strong log-concavity should be expected to hold under typical circumstances. Both Skalse & Abate (2023) and Hong et al. (2022) recognise these issues as limitations that should be lifted in future work. The analysis we carry out in this paper is not subject to any of these limitations. Moreover, in addition to being based on a more sound problem formulation, our paper also contains several novel results that are not analogous to any results derived by Skalse & Abate (2023) or Hong et al. (2022).

There are also earlier papers that study some specific types of misspecification in IRL. In particular, Freedman et al. (2020) study the effects of misspecified *choice sets* in IRL, and show that such misspecification in some cases can be catastrophic, and Viano et al. (2021) study the effects of misspecified *environment dynamics*, and propose an algorithm for reducing this effect. By contrast, we present a broader analysis that covers *all* forms of misspecification, within a single framework. Also relevant is the work by Armstrong & Mindermann (2018), who show that it is impossible to simultaneously learn a reward function and a behavioural model from a single data set, given an inductive bias towards joint simplicity.

---

[1]Indeed, there are detectable differences between data collected from human subjects and data synthesised using these standard behavioural models, see Orsini et al. (2021).

## 1.2 Preliminaries

A *Markov Decision Processes* (MDP) is a tuple $(\mathcal{S}, \mathcal{A}, \tau, \mu_0, R, \gamma)$ where $\mathcal{S}$ is a set of *states*, $\mathcal{A}$ is a set of *actions*, $\tau : \mathcal{S} \times \mathcal{A} \to \Delta(\mathcal{S})$ is a *transition function*, $\mu_0 \in \Delta(\mathcal{S})$ is an *initial state distribution*, $R : \mathcal{S} \times \mathcal{A} \times \mathcal{S} \to \mathbb{R}$ is a *reward function*, and $\gamma \in (0, 1)$ is a *discount rate*. Here $\Delta(X)$ denotes the set of all probability distributions over $X$. In this paper, we assume that $\mathcal{S}$ and $\mathcal{A}$ are finite, and that all states are reachable under $\tau$ and $\mu_0$. A *policy* is a function $\pi : \mathcal{S} \to \Delta(\mathcal{A})$. A *trajectory* $\xi = \langle s_0, a_0, s_1, a_1 \dots \rangle$ is a possible path in an MDP. The *return function* $G$ gives the cumulative discounted reward of a trajectory, $G(\xi) = \sum_{t=0}^{\infty} \gamma^t R(s_t, a_t, s_{t+1})$, and the *evaluation function* $J$ gives the expected trajectory return given a policy, $J(\pi) = \mathbb{E}_{\xi \sim \pi}[G(\xi)]$. A policy maximising $J$ is an *optimal policy*. The *value function* $V^\pi : \mathcal{S} \to \mathbb{R}$ of a policy encodes the expected future discounted reward from each state when following that policy, and the $Q$-function is $Q^\pi(s, a) = \mathbb{E}[R(s, a, S') + \gamma V^\pi(S')]$. $Q^\star$ and $V^\star$ denote the optimal $Q$- and value-function.

An IRL algorithm needs a *behavioural model* that describes how the observed policy $\pi$ relates to the underlying reward function $R$. In the current IRL literature, the most common models are *optimality*, where it is assumed that $\pi$ is optimal under $R$ (e.g. Ng & Russell, 2000), *Boltzmann-rationality*, where it is assumed that $\mathbb{P}(\pi(s) = a) \propto e^{\beta Q^\star(s,a)}$, where $\beta$ is a temperature parameter (e.g. Ramachandran & Amir, 2007), and *maximal causal entropy (MCE)*, where it is assumed that $\pi$ maximises the causal entropy objective, which is given by $\mathbb{E}[\sum_{t=0}^{\infty} \gamma^t (R(S_t, A_t, S_{t+1}) + \alpha H(\pi(S_t)))]$, where $\alpha$ is a weight and $H$ is the Shannon entropy function (e.g. Ziebart, 2010).

Two reward functions $R_1, R_2$ are said to differ by *potential shaping* (with $\gamma$) if there is a function $\Phi : \mathcal{S} \to \mathbb{R}$ such that $R_2(s, a, s') = R_1(s, a, s') + \gamma \Phi(s') - \Phi(s)$ (Ng et al., 1999), by $S'$-*redistribution* (with $\tau$) if $\mathbb{E}_{S' \sim \tau(s,a)}[R_1(s, a, S')] = \mathbb{E}_{S' \sim \tau(s,a)}[R_2(s, a, S')]$ (Skalse et al., 2022), and by *positive linear scaling* if there is a positive constant $c$ such that $R_2 = c \cdot R_1$.

Given a set $X$, a *pseudometric* on $X$ is any function $d : X \times X \to \mathbb{R}$ that satisfies the conditions that $d(x, x) = 0$, $d(x, y) \geq 0$, $d(x, y) = d(y, x)$, and $d(x, z) \leq d(x, y) + d(y, z)$, for all $x, y, z \in X$. A *metric* additionally satisfies that if $d(x, y) = 0$ then $x = y$.

If a reward function $R$ satisfies that $J(\pi_1) = J(\pi_2)$ for all $\pi_1$ and $\pi_2$, then we say that $R$ is *trivial*. All constant reward functions are trivial, but there are always non-constant trivial rewards as well.[2]

## 2 Theoretical Framework

In this section, we will introduce the theoretical definitions and machinery that we will later use to analyse how robust the IRL problem is to different forms of misspecification.

### 2.1 Defining Misspecification Robustness

Before we can derive formal results about what types of misspecification different IRL algorithms are robust to, we first need to construct an abstract model of the IRL problem. First of all, assume that we have a fixed set of states $\mathcal{S}$ and a fixed set of actions $\mathcal{A}$, that $\mathcal{R}$ is the set of all reward functions $R : \mathcal{S} \times \mathcal{A} \times \mathcal{S} \to \mathbb{R}$, and that $\Pi$ is the set of all policies $\pi : \mathcal{S} \to \Delta(\mathcal{A})$. We say that a *behavioural model* is a function $f : \mathcal{R} \to \Pi$, i.e. a function that takes a reward function and returns a policy. For example, the function that, given $R$, returns the Boltzmann-rational policy of $R$ under transition function $\tau$, discount $\gamma$, and temperature $\beta$, is an example of a behavioural model. Using this, we can now model the IRL learning problem as follows: first, we assume that there is a true underlying reward function $R^\star$, and that the training data is generated by a behavioural model $g$, so that the learning algorithm observes the policy $\pi$ given by $g(R^\star)$. Moreover, we assume that an IRL algorithm $\mathcal{L}$ has a model $f$ of how the observed policy $\pi$ relates to $R^\star$, where $f$ is also a behavioural model, such that $\mathcal{L}$ converges to a reward function $R_h$ which satisfies $f(R_h) = \pi = g(R^\star)$. If $f \neq g$, then $f$ is *misspecified*, and otherwise $f$ is correctly specified.

For convenience, if $f(R_1) = f(R_2)$ whenever $R_1$ and $R_2$ differ by potential shaping, then we say that $f$ is *invariant to potential shaping*, and similarly for $S'$-redistribution and positive linear scaling.

---

[2]These are given by applying potential shaping and $S'$-redistribution to a constant reward function.

In many contexts, we have prior information about $R^\star$, beyond the information provided by $g(R^\star)$. For example, we may know that $R^\star$ is defined in terms of some sparse state features, or we may know that it only depends on the states, and so on. To model this, we give our definitions relative to an arbitrary set of reward functions $\hat{\mathcal{R}} \subseteq \mathcal{R}$. We then assume that the true reward function $R^\star$ is contained in $\hat{\mathcal{R}}$, and that the learning algorithm $\mathcal{L}$ will learn a reward function $R_h$ that is also contained in $\hat{\mathcal{R}}$. Unless otherwise stated, our results apply for any choice of $\hat{\mathcal{R}}$ (including $\hat{\mathcal{R}} = \mathcal{R}$).

Intuitively speaking, we want to say that a behavioural model $f$ is robust to misspecification with $g$ if a learning algorithm that is based on $f$ is guaranteed to learn a reward function that is "close" to the true reward function if it is trained on data generated from $g$. To make this statement formal, we need a definition of what it means for two reward functions to be "close". In this paper, we assume that this is defined in terms of some pseudometric $d^{\mathcal{R}}$ on $\mathcal{R}$. In Section 2.2, we discuss how to choose this pseudometric. Unless otherwise stated, our results apply for any choice of pseudometric. Note also that while $d^{\mathcal{R}}$ of course may be a proper metric, we allow it to be a pseudometric since we may want to consider distinct reward functions to be equivalent. Using this, we can now give our formal definition of misspecification robustness:

**Definition 1.** Given a set of reward functions $\hat{\mathcal{R}} \subseteq \mathcal{R}$, a pseudometric $d^{\mathcal{R}}$ on $\hat{\mathcal{R}}$, and two behavioural models $f, g : \hat{\mathcal{R}} \to \Pi$, we say that $f$ is $\epsilon$-robust to misspecification with $g$ if each of the following conditions are satisfied:

1. If $f(R_1) = g(R_2)$ then $d^{\mathcal{R}}(R_1, R_2) \leq \epsilon$, for all $R_1, R_2 \in \hat{\mathcal{R}}$.

2. If $f(R_1) = f(R_2)$ then $d^{\mathcal{R}}(R_1, R_2) \leq \epsilon$, for all $R_1, R_2 \in \hat{\mathcal{R}}$.

3. $\mathrm{Im}(g) \subseteq \mathrm{Im}(f)$ on $\hat{\mathcal{R}}$.

4. There exists $R_1, R_2 \in \hat{\mathcal{R}}$ such that $f(R_1) \neq g(R_2)$.

Before moving on, let us explain each of these conditions intuitively. The first condition is saying that any learning algorithm $\mathcal{L}$ based on $f$ is guaranteed to learn a reward function that has a distance of at most $\epsilon$ to the true reward function when trained on data generated from $g$; this is the core property of misspecification robustness. Similarly, the second condition is saying that any learning algorithm $\mathcal{L}$ based on $f$ is guaranteed to learn a reward function that has a distance of at most $\epsilon$ to the true reward function *when there is no misspecification*; this condition is included to rule out certain pathological edge cases. The third condition is effectively saying that $\mathcal{L}$ can never observe a policy that is impossible according to its model. Depending on how $\mathcal{L}$ behaves, it may in some cases be possible to drop this condition, but we include it to make our analysis as general as possible. The fourth condition is simply saying that $f$ and $g$ are distinct on $\hat{\mathcal{R}}$ (otherwise $f$ is not misspecified!). More extensive discussion of Definition 1, including more subtle issues, is given in Appendix A.

We are particularly interested in the three behavioural models that are most common in the current IRL literature, and so we will use special notation for these models. Given a transition function $\tau$ and a discount parameter $\gamma$, let $b_{\tau, \gamma, \beta} : \mathcal{R} \to \Pi$ be the function that returns the Boltzmann-rational policy of $R$ with temperature $\beta$, and let $c_{\tau, \gamma, \alpha} : \mathcal{R} \to \Pi$ be the function that returns the MCE policy of $R$ with weight $\alpha$. Similarly, let $o_{\tau, \gamma} : \mathcal{R} \to \Pi$ be the function that returns the optimal policy of $R$ that takes all optimal actions with equal probability.

## 2.2 Reward Function Metrics

We wish to obtain results that describe how different the learnt reward function $R_h$ may be compared to the underlying true reward function $R^\star$, given different forms of misspecification. To do this, we need a way to *quantify* the difference between $R_h$ and $R^\star$, in the form of a pseudometric $d^{\mathcal{R}}$ on $\mathcal{R}$. However, finding an appropriate choice of $d^{\mathcal{R}}$ is not straightforward. For example, suppose we simply let $d^{\mathcal{R}}(R_h, R^\star) = ||R_h - R^\star||_2$. In that case, we would have that $d^{\mathcal{R}}(R_h, R^\star)$ can be arbitrarily *large*, even if $R_h$ and $R^\star$ have the *same* ordering of policies, and similarly, that $d^{\mathcal{R}}(R_h, R^\star)$ can be arbitrarily *small*, even if $R_h$ and $R^\star$ have the *opposite* ordering of policies.[3] This means that the $\ell^2$-distance between $R_h$ and $R^\star$ does not quantify their qualitative difference in a useful way.

---

[3]To see this, let $R$ be any nontrivial reward function. Then for any positive $c$, we have that $R$ and $cR$ have the same ordering of policies, but by making $c$ large, we can make $||R - cR||_2$ arbitrarily large. Similarly, for

Intuitively, we want a pseudometric $d^{\mathcal{R}}$ with the property that $d^{\mathcal{R}}(R_h, R^\star)$ is small if (and only if) it would be safe to optimise $R_h$ instead of $R^\star$. As we have just argued, the $\ell^2$-norm does *not* satisfy this condition. Instead, we will use a *STARC-metric*, introduced by Skalse et al. (2023):

**Definition 2.** Let $\tau$ be a transition function and $\gamma$ be a discount factor. Given a reward function $R$, let $V_{\tau,\gamma}(R)$ be the set of all reward functions that differ from $R$ by potential shaping (with $\gamma$) and $S'$-redistribution (with $\tau$). Let $c_{\tau,\gamma}^{\mathrm{STARC}} : \mathcal{R} \to \mathcal{R}$ be the function where $c_{\tau,\gamma}^{\mathrm{STARC}}(R)$ returns the element of $V_{\tau,\gamma}(R)$ that has the smallest $\ell^2$-norm.[4] Moreover, let $s_{\tau,\gamma}^{\mathrm{STARC}} : \mathcal{R} \to \mathcal{R}$ be the function where $s_{\tau,\gamma}^{\mathrm{STARC}}(R) = c_{\tau,\gamma}^{\mathrm{STARC}}(R)/||c_{\tau,\gamma}^{\mathrm{STARC}}(R)||_2$ if $||c_{\tau,\gamma}^{\mathrm{STARC}}(R)||_2 > 0$, and $c_{\tau,\gamma}^{\mathrm{STARC}}(R)$ otherwise. Finally, let $d_{\tau,\gamma}^{\mathrm{STARC}} : \mathcal{R} \times \mathcal{R} \to [0, 1]$ be the function given by

$$d_{\tau,\gamma}^{\mathrm{STARC}}(R_1, R_2) = 0.5 \cdot ||s_{\tau,\gamma}^{\mathrm{STARC}}(R_1) - s_{\tau,\gamma}^{\mathrm{STARC}}(R_2)||_2.$$

There is extensive justification for measuring the distance between reward functions in terms of $d_{\tau,\gamma}^{\mathrm{STARC}}$. However, because the full justification is quite long, and because this justification is not essential to understand most of our results, we have decided to provide it in Appendix B, instead of the main text. Appendix B also contains a less formal, more intuitive explanation of Definition 2. For now, it is sufficient to know that $d_{\tau,\gamma}^{\mathrm{STARC}}(R_1, R_2) = 0$ if and only if $R_1$ and $R_2$ induce the same ordering of policies under $\tau$ and $\gamma$. Moreover, while some of our results are given in terms of $d_{\tau,\gamma}^{\mathrm{STARC}}$, we also have many results that apply for any choice of pseudometric on $\mathcal{R}$. This will be clearly stated in each theorem.

Another prominent pseudometric on $\mathcal{R}$ is EPIC, proposed by Gleave et al. (2021). Appendix C puts forward reasons to not use EPIC for the kind of analysis that we undertake this paper.

## 2.3 BACKGROUND RESULTS

In this section, we give a few important results from earlier works that we will rely on throughout this paper, and which will also be helpful for contextualising our results. First, it is useful to know under what conditions two reward functions have the same ordering of policies:

**Proposition 1.** $(\mathcal{S}, \mathcal{A}, \tau, \mu_0, R_1, \gamma)$ and $(\mathcal{S}, \mathcal{A}, \tau, \mu_0, R_2, \gamma)$ *have the same ordering of policies if and only if $R_1$ and $R_2$ differ by potential shaping (with $\gamma$), $S'$-redistribution (with $\tau$), and positive linear scaling.*

For a proof of Proposition 1, see Skalse & Abate (2023) (their Theorem 2.6). Also recall that $d_{\tau,\gamma}^{\mathrm{STARC}}(R_1, R_2) = 0$ if and only if $R_1$ and $R_2$ induce the same ordering of policies. Next, it is also useful to know under what conditions $f(R_1) = f(R_2)$ when $f$ is the Boltzmann-rational model or the maximal causal entropy model:

**Proposition 2.** *For any $\tau$, $\gamma$, and $\beta$, we have that $b_{\tau,\gamma,\beta}$ and $c_{\tau,\gamma,\alpha}$ are invariant to potential shaping (with $\gamma$) and $S'$-redistribution (with $\tau$), and no other transformations.*

For a proof of Proposition 2, see Skalse et al. (2022) (their Theorem 3.3). Note that Propositions 1 and 2 together imply that any learning algorithm $\mathcal{L}$ that is based on either the Boltzmann-rational model or the MCE-model is guaranteed to learn a reward function $R_h$ that has the same ordering of policies as the true reward function $R^\star$ when there is no misspecification.

# 3 MISSPECIFICATION ROBUSTNESS

In this section, we provide our main results about the misspecification robustness of various behavioural models, including the behavioural models that are most common in the IRL literature. Section 3.1 is quite dense, but 3.2 and 3.3 both provide more intuitive takeaways.

## 3.1 NECESSARY AND SUFFICIENT CONDITIONS

Given a behavioural model $f$, it is desirable to have necessary and sufficient conditions that completely characterise when $f$ is robust to misspecification with $g$. Surprisingly, our first result shows

---

any positive $\epsilon$, we have that $\epsilon R$ and $-\epsilon R$ have the opposite ordering of policies. However, by making $\epsilon$ small, we can make $||\epsilon R - (-\epsilon R)||_2$ arbitrarily small.

[4]$V_{\tau,\gamma}(R)$ is an affine subspace of $\mathcal{R}$, so there is a unique element of $V_{\tau,\gamma}(R)$ that minimises the $\ell^2$-norm.

that if $f(R_1) = f(R_2) \implies d^{\mathcal{R}}(R_1, R_2) = 0$, then we can derive such necessary and sufficient conditions in a relatively straightforward way:

**Theorem 1.** *Let $\hat{\mathcal{R}}$ be a set of reward functions, let $f : \hat{\mathcal{R}} \to \Pi$ be a behavioural model, and let $d^{\mathcal{R}}$ be a pseudometric on $\hat{\mathcal{R}}$. Assume that $f(R_1) = f(R_2) \implies d^{\mathcal{R}}(R_1, R_2) = 0$ for all $R_1, R_2 \in \hat{\mathcal{R}}$. Then $f$ is $\epsilon$-robust to misspecification with $g$ (as defined by $d^{\mathcal{R}}$) if and only if $g = f \circ t$ for some $t : \hat{\mathcal{R}} \to \hat{\mathcal{R}}$ such that $d^{\mathcal{R}}(R, t(R)) \leq \epsilon$ for all $R \in \hat{\mathcal{R}}$, and such that $f \neq g$.*

Recall that $d_{\tau,\gamma}^{\text{STARC}}(R_1, R_2) = 0$ if and only if $R_1$ and $R_2$ induce the same ordering of policies. Thus, if our reward metric is $d_{\tau,\gamma}^{\text{STARC}}$, then Theorem 1 applies to any behavioural model $f$ for which $f(R_1) = f(R_2)$ implies that $R_1$ and $R_2$ have the same policy ordering. Moreover, also recall that both the Boltzmann-rational model and the MCE model satisfy this condition (Propositions 1 and 2). Thus, if we can find the set $T_\epsilon$ of all transformations $t : \hat{\mathcal{R}} \to \hat{\mathcal{R}}$ such that $d^{\mathcal{R}}(R, t(R)) \leq \epsilon$, then we can get the set of all behavioural models $g$ to which $f$ is $\epsilon$-robust to misspecification by simply composing $f$ with each element of $T_\epsilon$. We next derive $T_\epsilon$:

**Proposition 3.** *A transformation $t : \mathcal{R} \to \mathcal{R}$ satisfies that $d_{\tau,\gamma}^{\text{STARC}}(R, t(R)) \leq \epsilon$ for all $R \in \mathcal{R}$ if and only if $t$ can be expressed as $t_1 \circ \cdots \circ t_{n-1} \circ t_n \circ t_{n+1} \circ \cdots \circ t_m$ for some $n$ and $m$ where*

$$||R - t_n(R)||_2 \leq ||c_{\tau,\gamma}^{\text{STARC}}(R)||_2 \cdot \sin(2 \arcsin(\epsilon/2))$$

*for all $R$, and for all $i \neq n$ and all $R$, we have that $R$ and $t_i(R)$ differ by potential shaping (with $\gamma$), $S'$-redistribution (with $\tau$), or positive linear scaling.*

The statement of Proposition 3 is quite terse, so let us briefly unpack it. First of all, $d_{\tau,\gamma}^{\text{STARC}}$ is invariant to any transformation that preserves the policy ordering of the reward function, and these transformations are exactly those that can be expressed as a combination of potential shaping, $S'$-redistribution, and positive linear scaling. As such, we can apply an arbitrary number of such transformations. Moreover, we can also transform $R$ in any way that does not change the standardised reward function $s_{\tau,\gamma}^{\text{STARC}}(R)$ by more than $\epsilon$; this is equivalent to the stated condition on $t_n$. Note that $\sin(2 \arcsin(\epsilon/2)) \approx \epsilon$ for small $\epsilon$, so the right-hand side is approximately equal to $\epsilon \cdot ||c_{\tau,\gamma}^{\text{STARC}}(R)||_2$. However, also note that $||c_{\tau,\gamma}^{\text{STARC}}(R)||_2 \leq ||R||_2$. Intuitively speaking, a reward transformation satisfies the conditions given in Proposition 3 if it never changes the policy order of the reward function by a large amount.

Using this, we can now state necessary and sufficient conditions that completely characterise all types of misspecification that the Boltzmann-rational model and the MCE model will tolerate:

**Corollary 1.** *Let $\hat{\mathcal{R}}$ be a set of reward functions, $\tau$ be a transition function, $\gamma$ a discount factor, $\beta$ a temperature parameter, and $\alpha$ a weight parameter. Let $\hat{T}_\epsilon$ be the set of all functions $t : \mathcal{R} \to \mathcal{R}$ that satisfy Proposition 3, and additionally satisfy that $t(R) \in \hat{\mathcal{R}}$ for all $R \in \hat{\mathcal{R}}$. Then $b_{\tau,\gamma,\beta} : \hat{\mathcal{R}} \to \Pi$ is $\epsilon$-robust to misspecification with $g$ (as defined by $d_{\tau,\gamma}^{\text{STARC}}$) if and only if $g = b_{\tau,\gamma,\beta} \circ t$ for some $t \in \hat{T}_\epsilon$ such that $b_{\tau,\gamma,\beta} \neq g$, and $c_{\tau,\gamma,\alpha} : \hat{\mathcal{R}} \to \Pi$ is $\epsilon$-robust to misspecification with $g$ (as defined by $d_{\tau,\gamma}^{\text{STARC}}$) if and only if $g = c_{\tau,\gamma,\alpha} \circ t$ for some $t \in \hat{T}_\epsilon$ such that $c_{\tau,\gamma,\alpha} \neq g$.*

In principle, Corollary 1 completely describes the misspecification robustness of the Boltzmann-rational model and of the MCE model (for any $\hat{\mathcal{R}}$). However, the statement of Corollary 1 is rather opaque, and difficult to interpret qualitatively. For this reason, we will in the subsequent sections examine a few important special types of misspecification, and analyse them individually.

We should also briefly comment on the fact that Corollary 1 does not cover $o_{\tau,\gamma}$, i.e. the optimality model. The reason for this is that, unless $|\mathcal{S}| = 1$ and $|\mathcal{A}| = 2$, there are reward functions $R_1, R_2$ such that $o_{\tau,\gamma}(R_1) = o_{\tau,\gamma}(R_2)$, but $d_{\tau,\gamma}^{\text{STARC}}(R_1, R_2) > 0$. This ought to be intuitive: two reward functions can have the same optimal policies, but have different policy orderings. This means that Theorem 1 does not apply to $o_{\tau,\gamma}$ when $d^{\mathcal{R}} = d_{\tau,\gamma}^{\text{STARC}}$. Moreover:

**Proposition 4.** *Unless $|\mathcal{S}| = 1$ and $|\mathcal{A}| = 2$, then for any $\tau$ and any $\gamma$ there exists an $E > 0$ such that for all $\epsilon < E$, there is no behavioural model $g$ such that $o_{\tau,\gamma}$ is $\epsilon$-robust to misspecification with $g$ (as defined by $d_{\tau,\gamma}^{\text{STARC}}$).*

This is simply a consequence of the fact that the second condition of Definition 1 will be violated for any $\epsilon$ that is sufficiently small. An analogous result will hold for any behavioural model $f$ and any pseudometric $d^{\mathcal{R}}$ for which $f(R_1) = f(R_2) \implies\!\!\!\!\!/ \ \ d^{\mathcal{R}}(R_1, R_2) = 0$.

## 3.2 PERTURBATION ROBUSTNESS

It is interesting to know whether or not a behavioural model $f$ is robust to misspecification with any behavioural model $g$ that is "close" to $f$. But what does it mean for $f$ and $g$ to be "close"? One option is to say that $f$ and $g$ are close if they always produce similar policies. In this section, we will explore under what conditions $f$ is robust to such misspecification, and provide necessary and sufficient conditions. Our results are given relative to a pseudometric $d^{\Pi}$ on $\Pi$. For example, $d^{\Pi}(\pi_1, \pi_2)$ may be the $\ell^2$-distance between $\pi_1$ and $\pi_2$, or it may be the KL divergence between their trajectory distributions, or it may be the $\ell^2$-distance between their occupancy measures, etc. As usual, our results apply for any choice of $d^{\Pi}$ unless otherwise stated. We can now define a notion of a *perturbation* and a notion of *perturbation robustness*:

**Definition 3.** Let $\hat{\mathcal{R}}$ be a set of reward functions, let $f, g : \hat{\mathcal{R}} \to \Pi$ be two behavioural models, and let $d^{\Pi}$ be a pseudometric on $\Pi$. Then $g$ is a $\delta$-perturbation of $f$ if $g \neq f$ and for all $R \in \hat{\mathcal{R}}$ we have that $d^{\Pi}(f(R), g(R)) \leq \delta$.

**Definition 4.** Let $\hat{\mathcal{R}}$ be a set of reward functions, let $f : \hat{\mathcal{R}} \to \Pi$ be a behavioural model, let $d^{\mathcal{R}}$ be a pseudometric on $\hat{\mathcal{R}}$, and let $d^{\Pi}$ be a pseudometric on $\Pi$. Then $f$ is $\epsilon$-robust to $\delta$-perturbation if $f$ is $\epsilon$-robust to misspecification with $g$ (as defined by $d^{\mathcal{R}}$) for any behavioural model $g : \hat{\mathcal{R}} \to \Pi$ that is a $\delta$-perturbation of $f$ (as defined by $d^{\Pi}$) with $\mathrm{Im}(g) \subseteq \mathrm{Im}(f)$.

A $\delta$-perturbation of $f$ simply is any function that is similar to $f$ on all inputs, and $f$ is $\epsilon$-robust to $\delta$-perturbation if a small perturbation of the observed policy leads to a small error in the inferred reward function. It would be desirable for a behavioural model to be robust in this sense. To start with, this captures any form of misspecification that always leads to a small change in the final policy. Moreover, in practice, we can often not observe the exact policy of the demonstrator, and must instead approximate it from a number of samples. In this case, we should also expect to infer a policy that is a perturbation of the true policy. Before moving on, we need one more definition:

**Definition 5.** Let $\hat{\mathcal{R}}$ be a set of reward functions, let $f : \hat{\mathcal{R}} \to \Pi$ be a behavioural model, let $d^{\mathcal{R}}$ be a pseudometric on $\hat{\mathcal{R}}$, and let $d^{\Pi}$ be a pseudometric on $\Pi$. Then $f$ is $\epsilon/\delta$-separating if $d^{\mathcal{R}}(R_1, R_2) > \epsilon \implies d^{\Pi}(f(R_1), f(R_2)) > \delta$ for all $R_1, R_2 \in \hat{\mathcal{R}}$.

Intuitively speaking, $f$ is $\epsilon/\delta$-separating if reward functions that are far apart, are sent to policies that are far apart.[5] Using this, we can now state our main result for this section:

**Theorem 2.** *Let $\hat{\mathcal{R}}$ be a set of reward functions, let $f : \hat{\mathcal{R}} \to \Pi$ be a behavioural model, let $d^{\mathcal{R}}$ be a pseudometric on $\hat{\mathcal{R}}$, and let $d^{\Pi}$ be a pseudometric on $\Pi$. Then $f$ is $\epsilon$-robust to $\delta$-perturbation (as defined by $d^{\mathcal{R}}$ and $d^{\Pi}$) if and only if $f$ is $\epsilon/\delta$-separating (as defined by $d^{\mathcal{R}}$ and $d^{\Pi}$).*

We have thus obtained necessary and sufficient conditions that describe when a behavioural model is robust to perturbations — namely, it has to be the case that this behavioural model sends reward functions that are far apart, to policies that are far apart. This ought to be quite intuitive; if two policies are close, then perturbations may lead us to conflate them. To be sure that the learnt reward function is close to the true reward function, we therefore need it to be the case that policies that are close always correspond to reward functions that are close (or, conversely, that reward functions which are far apart correspond to policies which are far apart).

Our next question is, of course, whether or not the standard behavioural models are $\epsilon/\delta$-separating. Surprisingly, we will show that this is *not* the case, when the distance between reward functions is measured using $d^{\mathrm{STARC}}_{\tau, \gamma}$, and the policy metric $d^{\Pi}$ is similar to Euclidean distance. Moreover, we only need very mild assumptions about the behavioural model to obtain this result:

**Theorem 3.** *Let $d^{\mathcal{R}}$ be $d^{\mathrm{STARC}}_{\tau, \gamma}$, and let $d^{\Pi}$ be a pseudometric on $\Pi$ which satisfies the condition that for all $\delta$ there exists a $\delta'$ such that if $||\pi_1 - \pi_2||_2 < \delta'$ then $d^{\Pi}(\pi_1, \pi_2) < \delta$. Let $c$ be any positive constant, and let $\hat{\mathcal{R}}$ be a set of reward functions such that if $||R||_2 = c$ then $R \in \hat{\mathcal{R}}$. Let $f : \hat{\mathcal{R}} \to \Pi$ be a behavioural model that is continuous. Then $f$ is not $\epsilon/\delta$-separating for any $\epsilon < 1$ or $\delta > 0$.*

---

[5]Note that this definition is *not* saying that reward functions which are close must be sent to policies which are close. In other words, $f$ being $\epsilon/\delta$-separating is *not* a continuity condition. It is also not a local property of $f$, but rather, a global property. It is, however, a continuity condition on the inverse of $f$.

This theorem is telling us several things at once. To make things easy, we can begin by noting that we may let $\hat{\mathcal{R}} = \mathcal{R}$, and that we may assume that $d^\Pi$ simply is the $\ell^2$-norm, i.e. $d^\Pi(\pi_1, \pi_2) = ||\pi_1 - \pi_2||_2$. Theorem 3 is then telling us that *no continuous behavioural model* is $\epsilon/\delta$-separating for any $\epsilon$ or $\delta$ (and therefore, by Theorem 2, also not $\epsilon$-robust to $\delta$-perturbation for any $\epsilon$ or $\delta$). Note that the Boltzmann-rational model and the maximal causal entropy model (i.e. $b_{\tau,\gamma,\beta}$ and $c_{\tau,\gamma,\alpha}$) both are continuous, and hence subject to Theorem 3. The condition given on $d^\Pi$ in Theorem 3 is simply a generalisation, that covers other policy-metrics than the $\ell^2$-norm.[6] Similarly, the condition on $\hat{\mathcal{R}}$ is also a generalisation to certain restricted reward spaces. We give a more in-depth, intuitive interpretation of Theorem 3, and an explanation of why it is true, in Appendix D.

### 3.3 MISSPECIFIED PARAMETERS

A behavioural model is typically defined relative to some parameters. For example, the Boltzmann-rational model is defined relative to a temperature parameter $\beta$ and a discount parameter $\gamma$, as well as the transition dynamics $\tau$. Moreover, determining the exact values of these parameters ex post facto can often be quite difficult. For example, there is a sizeable literature that attempts to estimate the rate at which humans discount future reward, and there is a fairly large range in the estimates that this literature produces (e.g. Percoco & Nijkamp, 2009). It is therefore interesting to know to what extent a behavioural model is robust to misspecification of its parameters. If $\pi$ is Boltzmann-rational for discount parameter $\gamma_1$, but an IRL algorithm interprets it as being Boltzmann-rational for discount parameter $\gamma_2$, where $\gamma_1 \approx \gamma_2$, then are we still guaranteed to learn a reward function that is close to the true reward function? These are the questions that we will study in this section.

We will first consider the case when the discount parameter, $\gamma$, is misspecified. Say that a transition function $\tau$ is *trivial* if for all states $s$ and all actions $a_1, a_2$, we have that $\tau(s, a_1) = \tau(s, a_2)$. We now have the following rather surprising result:

**Theorem 4.** *If $f_\gamma : \mathcal{R} \to \Pi$ is invariant to potential shaping with $\gamma$, and $\gamma_1 \neq \gamma_2$, then $f_{\gamma_1}$ is not $\epsilon$-robust to misspecification with $f_{\gamma_2}$ under $d^{\mathrm{STARC}}_{\tau,\gamma_3}$ for any non-trivial $\tau$, any $\gamma_3$, and any $\epsilon < 0.5$.*

Note that Theorem 4 permits that $\gamma_3 = \gamma_1$ or $\gamma_3 = \gamma_2$. Of course, any interesting environment will have a non-trivial transition function, so this requirement is very mild. Moreover, a $d^{\mathrm{STARC}}_{\tau,\gamma}$-distance of 0.5 is very large; this corresponds to the case where the reward functions are nearly orthogonal. This means that Theorem 4 is saying that if a behavioural model $f$ is invariant to potential shaping, then it is not robust to any misspecification of the discount parameter. Note that this holds even if $\gamma_1$ and $\gamma_2$ are arbitrarily close! Moreover, optimal policies, Boltzmann-rational policies, and MCE policies are all invariant to potential shaping, and hence $o_{\tau,\gamma}$, $b_{\tau,\gamma,\beta}$, and $c_{\tau,\gamma,\alpha}$ are subject to Theorem 4. In general, we should expect any behavioural model that uses exponential discounting to be invariant to potential shaping, and so Theorem 4 will apply very widely.

We will next consider the case when the transition function, $\tau$, is misspecified. Here, we similarly find that a very wide class of behavioural models are non-robust to any amount of misspecification:

**Theorem 5.** *If $f_\tau : \mathcal{R} \to \Pi$ is invariant to $S'$-redistribution with $\tau$, and $\tau_1 \neq \tau_2$, then $f_{\tau_1}$ is not $\epsilon$-robust to misspecification with $f_{\tau_2}$ under $d^{\mathrm{STARC}}_{\tau_3,\gamma}$ for any $\tau_3$, any $\gamma$, and any $\epsilon < 0.5$.*

Note that Theorem 5 permits that $\tau_3 = \tau_1$ or $\tau_3 = \tau_2$. Thus, Theorem 5 is saying that if a behavioural model $f$ is invariant to $S'$-redistribution, then it is not robust to any degree of misspecification of $\tau$ (even if $\tau_1$ and $\tau_2$ are arbitrarily close). Moreover, optimal policies, Boltzmann-rational policies, and maximal causal entropy policies, are all invariant to $S'$-redistribution, and hence $o_{\tau,\gamma}$, $b_{\tau,\gamma,\beta}$, and $c_{\tau,\gamma,\alpha}$ are subject to Theorem 5. Indeed, since $S'$-redistribution does not change the expected value of any policy, we should expect almost all sensible behavioural models to be invariant to $S'$-redistribution. As such, Theorem 5 will also apply very widely.

Theorem 4 and 5 show that a very wide range of behavioural models in principle are highly sensitive to arbitrarily small misspecification of two of their core parameters. To make this result more accessible and easier to understand, we have included two examples in Appendix E that explain the intuition behind these two theorems.

---

[6]Note that while Theorem 3 uses a "special" pseudometric on $\mathcal{R}$, in the form of $d^{\mathrm{STARC}}_{\tau,\gamma}$, we do not need to use a special (pseudo)metric on $\Pi$, because for policies, $\ell^2$ does capture the relevant notion of similarity.

Before moving on, we also want to note that the Boltzmann-rational model is robust to arbitrary misspecification of the temperature parameter, $\beta$, and that the maximal causal entropy model is robust to arbitrary misspecification of the weight parameter, $\alpha$. This was shown by Skalse & Abate (2023), in their Theorems 3.2 and 3.4. Specifically, we have that for any $\tau$ and $\gamma$, any $\epsilon \geq 0$, and any $\beta_1, \beta_2, \alpha_1, \alpha_2$, we have that $b_{\tau,\gamma,\beta_1}$ is $\epsilon$-robust to misspecification with $b_{\tau,\gamma,\beta_2}$, and that $c_{\tau,\gamma,\alpha_1}$ is $\epsilon$-robust to misspecification with $c_{\tau,\gamma,\alpha_2}$, as defined by $d_{\tau,\gamma}^{\mathrm{STARC}}$. For a detailed description of how to connect the results of Skalse & Abate (2023) to ours, see Appendix G.

## 4    DISCUSSION

We have quantified how robust IRL is to misspecification of the behavioural model. We first provided necessary and sufficient conditions that fully describe what types of misspecification many behavioural models will tolerate. In principle, these conditions give a complete answer to how tolerant a given behavioural model is to any given type of misspecification. However, these conditions are rather opaque, and difficult to interpret. Therefore, we have also separately provided necessary and sufficient conditions that characterise when a behavioural model is robust to *perturbation*, and we have analysed how robust many behavioural models are to misspecification of the discount factor $\gamma$ or the environment dynamics $\tau$. Our analysis suggests that the IRL problem is highly sensitive to many plausible forms of misspecification. In particular, a very wide class of behavioural models are unable to guarantee robust inference under arbitrarily small perturbations of the observed policy, or under arbitrarily small misspecification of $\gamma$ or $\tau$.

Our results present a serious challenge to IRL in the context of preference elicitation. The relationship between human preferences and human behaviour is very complex, and while it is certainly possible to create increasingly accurate models of human behaviour, it will never be realistically possible to create a model that is completely free from all forms of misspecification. Therefore, if IRL is unable to guarantee accurate inferences under even mild misspecification of the behavioural model, as our results suggest, then we should expect it to be very difficult (and perhaps even prohibitively difficult) to guarantee that IRL reliably will produce accurate inferences in real-world situations. This in turn means that IRL should be used cautiously, and that the learned reward functions should be carefully examined and evaluated (as done by e.g. Michaud et al., 2020; Jenner & Gleave, 2022). It also means that we need IRL algorithms that are specifically designed to be more robust under misspecification, such as e.g. that proposed by Viano et al. (2021). It may also be fruitful to combine IRL with other data sources, as done by e.g. Ibarz et al. (2018), or consider policy optimisation algorithms that assume that the reward function may be misspecified, as done by e.g. Krakovna et al. (2018; 2020); Turner et al. (2020); Griffin et al. (2022).

We also need more extensive investigations into the issue of how robust IRL is to misspecification, and there are several ways that our analysis can be extended. First of all, it may in some cases be possible to mitigate some of our negative results if $\mathcal{R}$ is restricted. For example, Theorem 4 and 5 rely on the fact that we for any reward $R_1$ can find a reward $R_2$ such that $R_1$ and $R_2$ differ by potential shaping or $S'$-redistribution for a given choice of $\gamma$ and $\tau$, but such that $R_1$ and $R_2$ have a large STARC-distance for other choices of $\gamma$ and $\tau$. We may thus be able to circumvent these results by restricting $\hat{\mathcal{R}}$ in a way that removes all such reward pairs. However, this is of course not straightforward, not least because we need to ensure that the true reward in fact is contained in $\hat{\mathcal{R}}$. Moreover, some of our results rely on the fact that we use STARC-metrics to quantify the difference between reward functions. While there are compelling theoretical justifications for doing so (cf. Appendix B and C), there may be other relevant options. STARC-metrics are quite strong, and we may be able to derive weaker guarantees using other forms of reward metrics. Furthermore, it may also be fruitful to modify Definition 1, for example by making it more probabilistic, or generalising it in other ways. This topic is discussed in Appendix A. Finally, our analysis can also be extended to other types of behavioural models and other types of misspecification. For example, are policies that use (e.g.) *hyperbolic discounting* subject to a result that is analogous to Theorem 4? Such investigations also present an interesting direction for future work.

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

## A   MOTIVATING OUR DEFINITION OF MISSPECIFICATION ROBUSTNESS

In this section, we provide further discussion and motivation for our formalisation of misspecification robustness, given in Definition 1, beyond the discussion we give in Section 2.1.

### A.1   ADDITIONAL COMMENTS ON THE CONDITIONS FOR MISSPECIFICATION ROBUSTNESS

In this section, we make a few additional comments on some of the four conditions in Definition 1. In particular, while the first condition ought to be reasonably clear, we have further comments on each of the remaining three conditions.

Condition 2 says that for all $R_1, R_2 \in \hat{\mathcal{R}}$, if $f(R_1) = f(R_2)$ then $d^{\mathcal{R}}(R_1, R_2) \leq \epsilon$. In other words, any learning algorithm $\mathcal{L}$ based on $f$ is guaranteed to learn a reward function that has a distance of at most $\epsilon$ to the true reward function when trained on data generated by $f$, i.e. when there is no misspecification. It may not be immediately obvious why this assumption is included, since we assume that the data is generated by $g$, where $f \neq g$. To see this, suppose $\hat{\mathcal{R}} = \{R_1, R_2, R_3, R_4\}$ where $d^{\mathcal{R}}(R_1, R_2) < \epsilon$, $d^{\mathcal{R}}(R_3, R_4) < \epsilon$, and $d^{\mathcal{R}}(R_2, R_3) \gg \epsilon$, and let $f, g : \hat{\mathcal{R}} \to \Pi$ be two behavioural models where $f(R_1) = \pi_1$, $f(R_2) = f(R_3) = \pi_2$, $f(R_4) = \pi_3$, and $g(R_1) = g(R_2) = \pi_1$, $g(R_3) = g(R_4) = \pi_3$. This is illustrated in the diagram below:

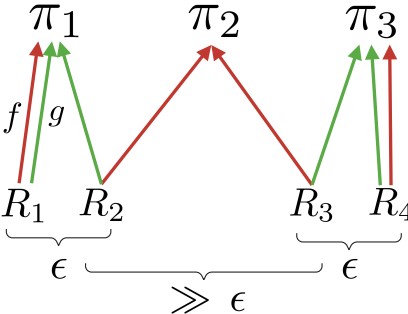

In this case, we have that $f(R_2) = f(R_3)$, but $d^{\mathcal{R}}(R_2, R_3) \gg \epsilon$. As such, $f$ violates condition 2 in Definition 1; a learning algorithm $\mathcal{L}$ based on $f$ is *not* guaranteed to learn a reward function that has distance at most $\epsilon$ to the true reward function when there is no misspecification, because $f$ cannot distinguish between $R_2$ and $R_3$, which have a large distance. However, if $f(R) = g(R')$, it does in this case follow that $d^{\mathcal{R}}(R, R') \leq \epsilon$. In other words, if the training data is coming from $g$, then a learning algorithm $\mathcal{L}$ based on $f$ *is* guaranteed to learn a reward function that has distance at most $\epsilon$ to the true reward function. As such, we could define misspecification robustness in such a way that $f$ would be considered to be robust to misspecification with $g$ in this case. However, this seems unsatisfactory, because $g$ essentially has to be carefully designed specifically to avoid certain blind spots in $f$. In other words, while condition 1 in Definition 1 is met, it is only met *spuriously*. To rule out these kinds of edge cases, we have therefore included the condition that for all $R_1, R_2 \in \hat{\mathcal{R}}$, if $f(R_1) = f(R_2)$, then it must be that $d^{\mathcal{R}}(R_1, R_2) \leq \epsilon$.

Condition 3 says that there for all $R_1 \in \hat{\mathcal{R}}$ exists an $R_2 \in \hat{\mathcal{R}}$ such that $f(R_2) = g(R_1)$. Stated differently, the image of $g$ on $\hat{\mathcal{R}}$ is a subset of the image of $f$ on $\hat{\mathcal{R}}$. The reason for why this assumption is necessary is to ensure that the learning algorithm can never observer data that is impossible according to its assumed model. For example, suppose $f$ maps each reward function to a deterministic policy; in that case, the learning algorithm $\mathcal{L}$ will assume that the observed policy must be deterministic. What happens if such an algorithm is given data from a nondeterministic policy? This is undefined, absent further details about $\mathcal{L}$, because $\mathcal{L}$ cannot possibly find a reward function that fits the training data under its assumed model. Since we do not want to make any strong assumptions about $\mathcal{L}$, it therefore seems reasonable to say that if $f$ always produces a deterministic policy, and $g$ sometimes produces nondeterministic policies, then $f$ is not robust to misspecification with $g$. More generally, it has to be the case that any policy that could be produced by $g$, can be

explained under $f$. This is encompassed by the condition that there for all $R_1 \in \hat{\mathcal{R}}$ exists an $R_2 \in \hat{\mathcal{R}}$ such that $f(R_2) = g(R_1)$. Of course, in many cases we may have that $\text{Im}(f) = \Pi$, i.e. that $f$ can produce any policy, and in that case this condition is vacuous.

Condition 4 says that there exists $R_1, R_2 \in \hat{\mathcal{R}}$ such that $f(R_1) \neq f(R_2)$; in other words, that $f \neq g$ on $\hat{\mathcal{R}}$. This condition is not strictly necessary – from a mathematical standpoint, very little would change if we were to simply remove it from Definition 1. Indeed, the only effect that this condition has on the results in this paper is that Theorem 1 and Corollary 1 add the condition that $f \neq g$, and that $f \neq g$ is part of the definition of $\delta$-perturbations (Definition 3). Rather, the reason for including the assumption that $f \neq g$ is purely to make Definition 1 more intuitive. If $f = g$, then $f$ is not misspecified, and it would seem odd to say that "$f$ is $\epsilon$-robust to misspecification with itself". As such, there is no deeper significance to this condition besides making our terminology more clear.

## A.2 On the Assumption That Behavioural Models Are Functions

Here, we will comment on the fact that behavioural models are assumed to be *functions*; i.e., we assume that a behavioural model associate each reward function $R$ with a unique policy $\pi$. This is true for the Boltzmann-rational model and the maximal causal entropy model, but it may not be a natural assumption in all cases. For example, there may in general be more than one optimal policy. Thus, an optimal agent could associate some reward functions $R$ with multiple policies $\pi$. This particular example is not too problematic, because the set of all optimal policies still form a convex set. As such, it is natural to assume that an optimal agent would take all optimal actions with equal probability, which is what we have done in the definition of $o_{\tau,\gamma}$.[7] However, we could imagine alternative criteria which would associate some rewards with multiple policies, and where there may not be any canonical way to select a single policy among them. Such criteria may then not straightforwardly translate into a *functional* behavioural model.

There are several ways to handle such cases within our framework. First of all, we may simply assume that the observed agent still has some fixed method for breaking ties between policies that it considers to be equivalent (as we do for $o_{\tau,\gamma}$). In that case, we still ultimately end up with a function from $\mathcal{R}$ to $\Pi$, in which case our framework can be applied without modification. We expect this approach to be satisfactory in most cases.

It is worth noting that this approach does not necessarily require us to actually know how the observed agent breaks ties between equivalent policies. To see this, let $G : \mathcal{R} \to \mathcal{P}(\Pi)$ be a function that associates each reward function with a set of policies. We can then say that a behavioural model $g : \mathcal{R} \to \Pi$ *implements* $G$ if $g(R) \in G(R)$ for all $R \in \mathcal{R}$. Using this definition, we could then say that $f : \mathcal{R} \to \Pi$ is robust to misspecification with $G : \mathcal{R} \to \mathcal{P}(\Pi)$ if $f$ is robust to misspecification with each $g$ that implements $G$, where $f$ being robust to misspecification with $g$ is defined as in Definition 1. In other words, we assume that the observed agent has a fixed method for breaking ties between policies in $G$, but without making any assumptions about what this method is. Using that definition, our framework can then be applied without modification.

An alternative approach could be to generalise the definition of behavioural models to allow them to return a set of policies, i.e. $f : \mathcal{R} \to \mathcal{P}(\Pi)$. Most of our results can be extended to cover this case in a mostly straightforward manner. However, this approach is somewhat unsatisfactory, because we would then assume that the learning algorithm $\mathcal{L}$ gets to observe all policies in the set $f(R^\star)$. However, in reality, it seems more realistic to assume that $\mathcal{L}$ only gets to observe a single element of $f(R^\star)$, unless perhaps $\mathcal{L}$ gets data from multiple similar agents acting in the same environment.

## A.3 On Restricted Spaces of Reward Functions

Our definitions are given relative to a set of reward functions $\hat{\mathcal{R}}$, which in general may be any subset of $\mathcal{R}$. It may not be immediately obvious why this is necessary, and so we will say a few words about that issue here.

---

[7]Note also that this is equivalent to assuming that an optimal agent would take all optimal actions with *positive* probability, but that the exact probability that it associates with each action does not convey any further information about $R$.

First of all, we should note that we always allow $\hat{\mathcal{R}} = \mathcal{R}$. This means that the introduction of $\hat{\mathcal{R}}$ makes our analysis strictly more general, in the sense that we always can assume that $\hat{\mathcal{R}}$ is unrestricted. In other words, nothing is lost by giving our definitions and theorem statements relative to a set of reward functions $\hat{\mathcal{R}}$, instead of the set of all reward functions.

Moreover, there are many cases where it is interesting to restrict $\mathcal{R}$. To start with, we use reward functions with type signature $\mathcal{S} \times \mathcal{A} \times \mathcal{S} \to \mathbb{R}$, but it is quite common to use reward functions with a different type signature, such as for example $\mathcal{S} \times \mathcal{A} \to \mathbb{R}$ or $\mathcal{S} \to \mathbb{R}$. We can ensure that our analysis covers these settings as well, by noting that we can allow $\hat{\mathcal{R}}$ to be equal to $\{R \in \mathcal{R} \mid \forall s, a, s_1, s_2 : R(s, a, s_1) = R(s, a, s_2)\}$, or $\{R \in \mathcal{R} \mid \forall s, a_1, a_2, s_1, s_2 : R(s, a_1, s_1) = R(s, a_2, s_2)\}$, and so on. As such, by using a (potentially restricted) set of rewards $\hat{\mathcal{R}}$, we can make sure that our results do not depend on these design choices.

Additionally, there are many cases where we may have *prior information* about the underlying true reward function $R^\star$, over and above the information provided by the observed policy. For example, we may know that the reward function cannot depend on certain features of the environment, or we may know that it only depends on the state of the environment at the end of an episode, and so on. This information may come from expert knowledge, or from auxiliary data sources, etc. In these cases, it makes sense to restrict $\hat{\mathcal{R}}$ to the set of all reward functions that are viable in light of this prior knowledge. Moreover, restricted reward sets also allow us to handle the case where this information is given in the form of a Bayesian prior, see Appendix A.4.

Another reason for restricting $\mathcal{R}$ is that Definition 1 is existential, in the sense that a single counterexample in principle is enough to prevent $f$ from being $\epsilon$-robust to misspecification with $g$, even if $f(R_1) = g(R_2)$ implies $d^{\mathcal{R}}(R_1, R_2) \leq \epsilon$ for "most" $R_1$ and $R_2$, etc. As such, even if $f$ is not $\epsilon$-robust to misspecification with $g$, it could in theory still be the case that a learning algorithm $\mathcal{L}$ based on $f$ is guaranteed to learn a reward function $R_h$ that is close to the true reward function $R^\star$ for most choices of $R^\star$. We can rule out this possibility by restricting $\hat{\mathcal{R}}$ in a way that excludes gerrymandered counter-examples.

As such, by giving our definitions relative to a set of reward functions $\hat{\mathcal{R}}$, we make our analysis more versatile and more general.

## A.4 On Making the Analysis More Probabilistic

The formalisation of misspecification robustness in Definition 1 is essentially a worst-case analysis, in the sense that it requires each condition to hold for *all* reward functions. For example, a single pair of rewards $R_1$, $R_2$ with $f(R_1) = g(R_2)$ and $d^{\mathcal{R}}(R_1, R_2) > \epsilon$ is enough to make it so that $f$ is not $\epsilon$-robust to misspecification with $g$, even if $f(R_1) = g(R_2)$ implies $d^{\mathcal{R}}(R_1, R_2) \leq \epsilon$ for "most" reward functions. This makes sense if we do not want to make any assumptions about the true reward function $R^\star$, or about the inductive bias of the learning algorithm. However, in certain cases, we may know that $R^\star$ is sampled from a particular distribution $\mathcal{D}$ over $\mathcal{R}$. In those cases, it may be more relevant to know whether $d^{\mathcal{R}}(R^\star, R_h) \leq \epsilon$ with high probability.

To make this more formal, we may assume that we have two behavioural models $f, g : \mathcal{R} \to \Pi$ and a distribution $\mathcal{D}$ over $\mathcal{R}$, that $R^\star$ is sampled from $\mathcal{D}$, and that the learning algorithm $\mathcal{L}$ observes the policy $\pi = g(R^\star)$. We then assume that $\mathcal{L}$ returns the reward function $R_h$ such that $f(R_h) = \pi$, and that $\mathcal{L}$ selects among all such reward functions using some (potentially nondeterministic) inductive bias. We then want to know if $d^{\mathcal{R}}(R^\star, R_h) \leq \epsilon$ with probability at least $1 - \delta$, for some $\delta$ and $\epsilon$.

Our framework can, to an extent, be used to study this setting as well. In particular, suppose we pick a set $\hat{\mathcal{R}}$ of "likely" reward functions such that $\mathbb{P}_{R \sim \mathcal{D}}(R \in \hat{\mathcal{R}}) \geq 1 - \delta$, and such that the learning algorithm $\mathcal{L}$ will return a reward function $R_h \in \hat{\mathcal{R}}$ if there exists a reward function $R_h \in \hat{\mathcal{R}}$ such that $f(R_h) = g(R^\star)$. Then if $f$ is $\epsilon$-robust to misspecification with $g$ on $\hat{\mathcal{R}}$, we have that $\mathcal{L}$ will learn a reward function $R_h$ such that $d^{\mathcal{R}}(R^\star, R_h) \leq \epsilon$ with probability at least $1 - \delta$.

So, for example, suppose $\hat{\mathcal{R}}$ is the set of all reward functions that have "low complexity", for some complexity measure and complexity threshold. The above argument then informally tells us that if the true reward function is likely to have low complexity, and if $\mathcal{L}$ will attempt to fit a low-complexity reward function to its training data, then the learnt reward function will be close to the true reward

function with high probability, as long as $f$ is $\epsilon$-robust to misspecification with $g$ on the set of all low-complexity reward functions.

Thus, while Definition 1 gives us a worst-case formalisation of misspecification robustness, it is relatively straightforward to carry out a more probabilistic analysis within the same framework.

## B    EXPLAINING AND MOTIVATING STARC-METRICS

In this section, we will explain Definition 2, and provide the theoretical justification for measuring the difference between reward functions using $d_{\tau,\gamma}^{\mathrm{STARC}}$.

Let us first walk through the definition of $d_{\tau,\gamma}^{\mathrm{STARC}}$, and explain each of the steps. Intuitively speaking, we want to consider $R_1$ and $R_2$ to be equivalent if (and only if) they induce the same ordering of policies. Moreover, also recall that $R_1$ and $R_2$ have the same ordering of policies if and only if they differ by potential shaping, $S'$-redistribution, and positive linear scaling (see Proposition 1). For this reason, $d_{\tau,\gamma}^{\mathrm{STARC}}$ first *standardises* each reward function in a way that maps all equivalent rewards to a single representative in their respective equivalence class, before measuring their difference.

To do this, we first use $c_{\tau,\gamma}^{\mathrm{STARC}}$ to map all rewards that differ by potential shaping and $S'$-redistribution to a single representative. Note that for all $R$, the set of all rewards that differ from $R$ by potential shaping and $S'$-redistribution forms an affine subspace. This means that there is a well-defined "smallest" element of each such equivalence class, which is the reward function that $c_{\tau,\gamma}^{\mathrm{STARC}}$ returns. It is also worth noting that $c_{\tau,\gamma}^{\mathrm{STARC}}$ is an orthogonal linear transformation, that maps $\mathcal{R}$ to an $|\mathcal{S}|(|\mathcal{A}| - 1)$-dimensional linear subspace of $\mathcal{R}$.

After this, we *normalise* the resulting reward functions, by dividing them by their $\ell^2$-norm. This collapses positive linear scaling, which now means that $s_{\tau,\gamma}^{\mathrm{STARC}}(R_1) = s_{\tau,\gamma}^{\mathrm{STARC}}(R_2)$ if and only if $R_1$ and $R_2$ have the same ordering of policies. We then measure the distance between the resulting reward functions, and multiply this distance by $0.5$ to ensure that the resulting value is between $0$ and $1$. For more details, see Skalse et al. (2023).

To get an intuitive sense of how $d_{\tau,\gamma}^{\mathrm{STARC}}$ behaves, first note that $d_{\tau,\gamma}^{\mathrm{STARC}}$ is a pseudometric on $\mathcal{R}$. Moreover, as we have already alluded to, $d_{\tau,\gamma}^{\mathrm{STARC}}(R_1, R_2) = 0$ if and only if $R_1$ and $R_2$ induce the same ordering of policies under $\tau$ and $\gamma$. In addition to this, we also have that $d_{\tau,\gamma}^{\mathrm{STARC}}(R_1, R_2) = 1$ if and only if $R_1$ and $R_2$ induce the *opposite* ordering of policies under $\tau$ and $\gamma$. Furthermore, if $R_0$ is trivial and $R$ is non-trivial, then we have that $d_{\tau,\gamma}^{\mathrm{STARC}}(R, R_0) = 0.5$. More generally, if $R_1$ and $R_2$ are approximately orthogonal, then $d_{\tau,\gamma}^{\mathrm{STARC}}(R_1, R_2) \approx 0.5$. As such, $d_{\tau,\gamma}^{\mathrm{STARC}}$ gives each pair of reward functions $R_1, R_2$ a distance between $0$ and $1$, where a distance close to $0$ means that $R_1$ and $R_2$ have approximately the same policy order, a distance close to $1$ means that they have approximately the opposite policy order, and a distance close to $0.5$ means that they are approximately orthogonal. Almost all reward functions have a distance close to $0.5$.

In addition to this, $d_{\tau,\gamma}^{\mathrm{STARC}}$ induces an upper bound on worst-case regret. Specifically:

**Definition 6.** A pseudometric $d$ on $\mathcal{R}$ is *sound* if there exists a positive constant $U$, such that for any reward functions $R_1$ and $R_2$, if two policies $\pi_1$ and $\pi_2$ satisfy that $J_2(\pi_2) \geq J_2(\pi_1)$, then

$$J_1(\pi_1) - J_1(\pi_2) \leq U \cdot (\max_\pi J_1(\pi) - \min_\pi J_1(\pi)) \cdot d(R_1, R_2).$$

**Proposition 5.** $d_{\tau,\gamma}^{\mathrm{STARC}}$ *is sound.*

For a proof of Proposition 5, see Skalse et al. (2023). Before moving on, let us briefly unpack Definition 6. $J_1(\pi_1) - J_1(\pi_2)$ is the regret, as measured by $R_1$, of using policy $\pi_2$ instead of $\pi_1$. Division by $\max_\pi J_1(\pi) - \min_\pi J_1(\pi)$ normalises this quantity to lie between $0$ and $1$ (though the term is put on the right-hand side of the inequality, instead of being used as a denominator, in order to avoid division by zero when $R_1$ is trivial. The condition that $J_2(\pi_2) \geq J_2(\pi_1)$ says that $R_2$ prefers $\pi_2$ over $\pi_1$. Taken together, this means that a pseudometric $d$ on $\mathcal{R}$ is sound if $d(R_1, R_2)$ gives an upper bound on the (normalised) maximal regret that could be incurred under $R_1$ if an arbitrary policy $\pi_1$ is optimised to another policy $\pi_2$ according to $R_2$. Note that we, as a special case, may assume that $\pi_1$ is optimal under $R_1$, and that $\pi_2$ is optimal under $R_2$. Since $d_{\tau,\gamma}^{\mathrm{STARC}}$ is sound, it induces such a bound.

In addition to this, $d_{\tau,\gamma}^{\mathrm{STARC}}$ also induces a *lower* bound on worst-case regret. It may not be immediately obvious why this property is desirable. To see why this is the case, note that if a pseudometric $d$ on $\mathcal{R}$ does not induce a lower bound on worst-case regret, then there are reward functions that have a low regret, but large distance under $d$. This would in turn mean that $d$ is not tight, and that it should be possible to find a better way to measure the distance between reward functions. If a pseudometric induces a lower bound on regret, then these kinds of cases are ruled out. When a pseudometric has this property, we say that it is *complete*:

**Definition 7.** A pseudometric $d$ on $\mathcal{R}$ is *complete* if there exists a positive constant $L$, such that for any reward functions $R_1$ and $R_2$, there exists two policies $\pi_1$ and $\pi_2$ such that $J_2(\pi_2) \geq J_2(\pi_1)$ and

$$J_1(\pi_1) - J_1(\pi_2) \geq L \cdot (\max_\pi J_1(\pi) - \min_\pi J_1(\pi)) \cdot d(R_1, R_2),$$

and moreover, if $R_1$ and $R_2$ have the same policy order then $d(R_1, R_2) = 0$.

**Proposition 6.** $d_{\tau,\gamma}^{\mathrm{STARC}}$ *is complete.*

Note that if $R_1$ and $R_2$ have the same policy order and $\max_\pi J_1(\pi) - \min_\pi J_1(\pi) > 0$, then $d(R_1, R_2) = 0$; the last condition ensures that this also holds when $\max_\pi J_1(\pi) - \min_\pi J_1(\pi) = 0$. Intuitively, if $d$ is sound, then a small $d$ is *sufficient* for low regret, and if $d$ is complete, then a small $d$ is *necessary* for low regret. Soundness implies the absence of false positives, and completeness the absence of false negatives. As per Proposition 6, we have that $d_{\tau,\gamma}^{\mathrm{STARC}}$ is complete, and hence tight. For a proof, see Skalse et al. (2023).

Moreover, if a pseudometric is both sound and complete, then this implies that it, in a certain sense, is unique. Specifically:

**Proposition 7.** *If two pseudometrics $d_1$, $d_2$ on $\mathcal{R}$ are both sound and complete, then $d_1$ and $d_2$ are bilipschitz equivalent.*

For a proof, see Skalse et al. (2023). Note that this means that any pseudometric on $\mathcal{R}$ that is both sound and complete must be bilipschitz equivalent to $d_{\tau,\gamma}^{\mathrm{STARC}}$. As such, $d_{\tau,\gamma}^{\mathrm{STARC}}$ is a canonical pseudometric on $\mathcal{R}$, in the sense that a small $d_{\tau,\gamma}^{\mathrm{STARC}}$-distance is both necessary and sufficient for low worst-case regret, and that any other pseudometric on $\mathcal{R}$ with this property also must be equivalent to $d_{\tau,\gamma}^{\mathrm{STARC}}$. Therefore, we think it is justified to regard $d_{\tau,\gamma}^{\mathrm{STARC}}$ as the "right" way to quantify the difference between reward functions.

Recent literature has proposed other pseudometrics for quantifying the difference between reward functions, namely EPIC (Gleave et al., 2021) and DARD (Wulfe et al., 2022). However, these do not enjoy the same strong theoretical guarantees as $d_{\tau,\gamma}^{\mathrm{STARC}}$. In particular, they are neither sound nor complete in the sense of $d_{\tau,\gamma}^{\mathrm{STARC}}$. For more details, see Skalse et al. (2023).

## C  WHY NOT USE EPIC?

Many of our results are invariant to the choice of pseudometric on $\mathcal{R}$, but when we do have to pick a particular metric, we use $d_{\tau,\gamma}^{\mathrm{STARC}}$. Another prominent pseudometric on $\mathcal{R}$ is EPIC, which was first proposed by Gleave et al. (2021), and has since become the most widely used pseudometric on $\mathcal{R}$ (as judged by the number of citations at the time of writing). So why are we not using EPIC in this paper? There is a simple reason for this, namely that EPIC is sensitive to $S'$-redistribution. Specifically, for any reward function $R$ and any $\delta \in (0, 1]$ there exists two reward functions $R_1$, $R_2$ such that $R$, $R_1$, and $R_2$ differ by $S'$-redistribution, but such that the EPIC-distance between $R_1$ and $R_2$ is $1 - \delta$. In other words, starting from an arbitrary reward function $R$ and using only $S'$-redistribution, we can find reward functions whose EPIC-distance is arbitrarily close to 1.

This is problematic, because essentially any behavioural model of interest is invariant to $S'$-redistribution (including $o_{\tau,\gamma}$, $b_{\tau,\gamma,\beta}$, and $c_{\tau,\gamma,\alpha}$). This means that any such model will violate condition 2 in Definition 1 for all $\epsilon < 1$ when $d^{\mathcal{R}}$ is the EPIC pseudometric. Moreover, this also means that if $f$ is $\epsilon$-robust to misspecification with $g$ (as defined by the EPIC distance), and $g$ is invariant to $S'$-redistribution, then it must be the case that $\epsilon \geq 0.5$ (c.f. Lemma 1). Since an EPIC-distance of $0.5$ is very large, such results are essentially vacuous. In other words, the EPIC-pseudometric is too loose, and cannot be used to derive any non-trivial results within the setting that we are concerned with in this paper.

In addition to this, $d_{\tau,\gamma}^{\text{STARC}}$ also yields stronger theoretical guarantees than EPIC; see Appendix B and Skalse et al. (2023).

## D  WHY ARE CONTINUOUS MODELS NOT ROBUST TO PERTURBATIONS?

In this section, we give a more in-depth interpretation and explanation of Theorem 3.

Intuitively speaking, the fundamental reason that Theorem 3 holds is because there is a mismatch between $\ell^2$-distance and STARC-distance. In particular, if $f$ is continuous, then it must send reward functions that are close under the $\ell^2$-norm to policies that are close under the $\ell^2$-norm. However, there are reward functions that are close under the $\ell^2$-norm but have a large STARC distance. Hence, if $f$ is continuous then it will send some reward functions that are far apart under $d_{\tau,\gamma}^{\text{STARC}}$ (but close under $\ell^2$) to policies which are close (under $\ell^2$), which in turn means that $f$ is not $\epsilon/\delta$-separating.

To see this, let $R$ be an arbitrary non-trivial reward function, and let $\epsilon$ be any positive constant. We then have that $\epsilon \cdot R$ and $-\epsilon \cdot R$ have the opposite policy ordering, which means that $d_{\tau,\gamma}^{\text{STARC}}(\epsilon \cdot R, -\epsilon \cdot R) = 1$. However, by making $\epsilon$ small enough, we can ensure that the $\ell^2$-distance between $\epsilon \cdot R$ and $-\epsilon \cdot R$ is arbitrarily small, and hence that $d^{\Pi}(\epsilon \cdot R, -\epsilon \cdot R) < \delta$. Thus, we have two reward functions that have a large STARC-distance that are sent to policies that are close.

This example is not too concerning by itself, because it only demonstrates that we may run into trouble for reward functions that are very close to $0$, and we may expect such reward functions to be unlikely (both in the sense that the observed agent is unlikely to have such a reward function, and in the sense that the inductive bias of the learning algorithm is unlikely to generate such a hypothesis). It would therefore be natural to restrict $\hat{\mathcal{R}}$ in some way, for example by imposing a minimum size on the $\ell^2$-norm of all considered reward functions, or by supposing that they are normalised. However, Theorem 3 tells us that this will not work either: as long as there is some positive constant $c$ such that if $||R||_2 = c$ then $R \in \hat{\mathcal{R}}$, then we can always find reward functions $R_1, R_2$ such that their $\ell^2$-distance is small but $d_{\tau,\gamma}^{\text{STARC}}(R_1, R_2)$ is large. Theorem 3 thus applies very widely.

## E  WHY IS IRL SENSITIVE TO MISSPECIFIED PARAMETERS?

In this section, we give a more in-depth explanation of Theorems 4 and 5.

To start with, the reason that Theorem 4 is true is that we for any reward function $R_1$ can find a reward function $R_2$ such that $R_1$ and $R_2$ differ by potential shaping with $\gamma_1$, but such that $R_1$ and $R_2$ have a different policy ordering under $\gamma_2$ (when $\gamma_1 \neq \gamma_2$). To see this, consider a simple environment with three states $s_0, s_1, s_2$, where $s_0$ is the initial state, and where the agent can choose to either go directly from $s_0$ to $s_2$, or choose to first visit state $s_1$:

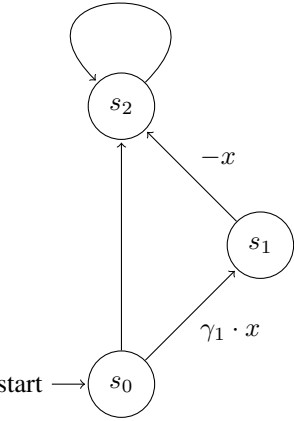

Let $R_1$ be any reward function over this environment, and let $R_2$ be the reward function that we get if we take $R_1$ and *increase* the reward of going from $s_0$ to $s_1$ by $\gamma_1 \cdot x$, and *decrease* the reward of

going from $s_1$ to $s_2$ by $x$. Now, the policy order under discounting with $\gamma_1$ is completely unchanged. At $s_1$, the value of every action is changed by the same amount, and so there is no reason to change action. Similarly, at $s_0$, the value of going to $s_1$ is changed by $\gamma_1 \cdot x - \gamma_1 \cdot x = 0$, and so there is likewise no reason to change action. This transformation corresponds to potential shaping where $\Phi(s_1) = x$ and $\Phi(s_0) = \Phi(s_2) = 0$. Therefore, if $f : \mathcal{R} \to \Pi$ is invariant to potential shaping with $\gamma_1$, then $f(R_1) = f(R_2)$.

However, if we discount with $\gamma_2$, then $R_1$ and $R_2$ have a different policy order. In particular, the value of going from $s_0$ to $s_1$ is changed by $\gamma_1 \cdot x - \gamma_2 \cdot x = (\gamma_1 - \gamma_2) \cdot x \neq 0$. Thus, if the optimal action under $R_1$ at $s_0$ is to go to $s_1$, then by making $x$ sufficiently large or sufficiently small (depending on whether $\gamma_1 > \gamma_2$, or vice versa), then we can create a reward function $R_2$ for which the optimal action instead is to go to $s_2$, and vice versa.

Thus, in this environment, for every reward function $R_1$ and every $\gamma_1, \gamma_2$ such that $\gamma_1 \neq \gamma_2$, we can find a reward function $R_2$ such that $R_1$ and $R_2$ differ by potential shaping with $\gamma_1$, but such that they have a different ordering of policies when we discount with $\gamma_2$. This in turn means that we cannot be robust to misspecification of $\gamma$; if the observed policy is computed using $\gamma_2$, then there are reward functions that would lead to the same observed policy (and which hence cannot be distinguished by the learning process) but which nonetheless are a large distance from each other as evaluated by $\gamma_1$. This issue is present as long as $\gamma_1 \neq \gamma_2$, and so the degree of misspecification does not matter.

This is the basic mechanism behind Theorem 4, although this theorem additionally shows that the dynamic which we describe above shows up for *any* non-trivial transition function. Intuitively speaking, we can use potential shaping to move reward around in the MDP (so that the agent receives a larger immediate reward at the cost of a lower reward later, or vice versa). However, because of the discounting, later rewards must be made larger than immediate rewards. If the discount values do not match, then this "compensation" will also not match, leading to a distortion of the policy ordering. Indeed, we can make it so that this distortion dominates the rest of the reward function. For the full details, see the proof of Theorem 4.

As for Theorem 4, this theorem is similarly true because we for any reward function $R_1$ can find a reward function $R_2$ such that $R_1$ and $R_2$ differ by $S'$-redistribution with $\tau_1$, but such that $R_1$ and $R_2$ have a different policy ordering under $\tau_2$ (when $\tau_1 \neq \tau_2$). To see this, suppose we have an MDP with (at least) three states $s_0, s_1, s_2$, and that taking action $a$ in state $s_0$ under transition function $\tau_1$ takes you to state $s_1$ with probability $p$, and $s_2$ with probability $1 - p$. Similarly, taking action $a$ in state $s_0$ under transition function $\tau_2$ takes you to state $s_1$ with probability $q$, and $s_2$ with probability $1 - q$, where $p \neq q$.

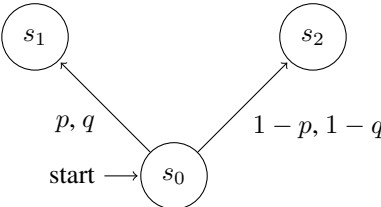

Let $R_1$ be any reward function, and $X$ any real number. Now note that we, regardless of the values of $R_1$ and $X$, can find values of $R_2(s_0, a, s_1)$ and $R_2(s_0, a, s_2)$ such that

$$p \cdot R_2(s_0, a, s_1) + (1 - p) \cdot R_2(s_0, a, s_2) = p \cdot R_1(s_0, a, s_1) + (1 - p) \cdot R_1(s_0, a, s_2)$$

and such that $q \cdot R_2(s_0, a, s_1) + (1 - q) \cdot R_2(s_0, a, s_2) = X$.

Note that this means that $\mathbb{E}_{S' \sim \tau_1(s_0, a)}[R_2(s_0, a, S')] = \mathbb{E}_{S' \sim \tau_1(s_0, a)}[R_1(s_0, a, S')]$, and that $\mathbb{E}_{S' \sim \tau_2(s_0, a)}[R_2(s_0, a, S')] = X$. Note also that $X$ was selected arbitrarily. In other words, the fact that $R_1$ and $R_2$ differ by $S'$-redistribution under $\tau_1$, leaves the expectation of $R_2$ under $\tau_2$ *completely unconstrained* for all transitions where $\tau_1 \neq \tau_2$. If $\tau_1 \neq \tau_2$ for all states, then the policy order of $R_2$ under $\tau_2$ can be literally any possible policy ordering. This in turn means that we cannot be robust to misspecification of $\tau$; if the observed policy is computed using $\tau_2$, then there are reward functions that would lead to the same observed policy (and which hence cannot be distinguished by the learning process) but which nonetheless have an arbitrarily large distance under $\tau_1$.

It is also important to note that Theorem 4 does not require that $\tau_1 \neq \tau_2$ for all states; indeed, it is enough for them to differ at just a single transition $s, a$. Using the same strategy as above, we can find two reward functions $R_1$ and $R_2$ such that $R_1$ and $R_2$ differ by $S'$-redistribution under $\tau_1$, but such that under $\tau_2$, the value of a given policy $\pi$ under $R_1$ depends primarily on visiting $s, a$ as many times as possible, but the value of $\pi$ under $R_2$ depends primarily on visiting $s, a$ as *few* times as possible. For the full details, see the proof of Theorem 4.

We can also give a somewhat less artificial example, to make this point more intuitive. Consider a simple $N \times N$ gridworld environment. We assume that the agent has four actions, up, down, left, and right. We assume that $\tau_1$ is deterministic, so that if the agent takes action up, then it moves one step up, etc. Moreover, we assume that $\tau_2$ is slippery, so that if the agent takes action up, then it moves up, up-left, and up-right with equal probability, and that if it takes action right, then it moves right, up-right, and down-right with equal probability, etc. For simplicity, we will also assume that the environment has a "PacMan-like" border, so that if the agent moves up from the top of the environment, then it ends up at the bottom, etc.[8]



Now suppose that $R_1$ and $R_2$ reward each transition depending on how the agent moves, according to the following schemas:

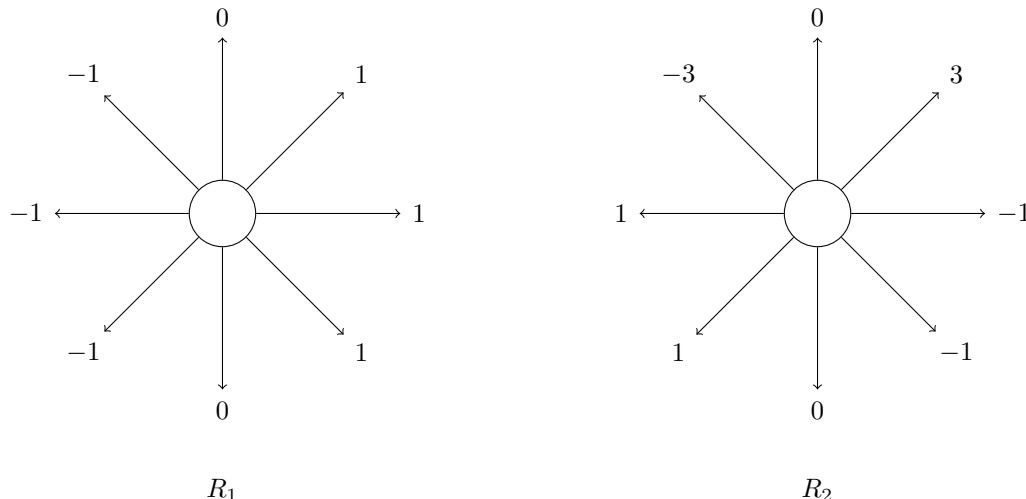

$$R_1 \qquad\qquad\qquad\qquad R_2$$

These two reward functions are identical under $\tau_2$, and give the agent 1 reward for going right, $-1$ for going left, and 0 for going up or down. However, under $\tau_1$, they are opposites; $R_1$ rewards the agent for going right, and $R_2$ rewards the agent for going left. Thus, if we observe a policy computed under $\tau_2$, then we will not be able to distinguish between $R_1$ and $R_2$, even though they have a large distance under $\tau_1$. Similar issues will occur given any discrepancy between $\tau_1$ and $\tau_2$.

---

[8]In other words, the environment is shaped like a torus.

# F    PROOFS

Here, we will provide the proofs of all our theorems and other theoretical results. The proofs are split up in three sections, mirroring the three subsections in Section 3.

## F.1    NECESSARY AND SUFFICIENT CONDITIONS

**Theorem 1.** *Let $\hat{\mathcal{R}}$ be a set of reward functions, let $f : \hat{\mathcal{R}} \to \Pi$ be a behavioural model, and let $d^{\mathcal{R}}$ be a pseudometric on $\hat{\mathcal{R}}$. Suppose that $f(R_1) = f(R_2) \implies d^{\mathcal{R}}(R_1, R_2) = 0$ for all $R_1, R_2 \in \hat{\mathcal{R}}$. Then $f$ is $\epsilon$-robust to misspecification with $g$ (as defined by $d^{\mathcal{R}}$) if and only if $g = f \circ t$ for some $t : \hat{\mathcal{R}} \to \hat{\mathcal{R}}$ such that $d^{\mathcal{R}}(R, t(R)) \leq \epsilon$ for all $R \in \hat{\mathcal{R}}$, and such that $f \neq g$.*

*Proof.* For the first direction, let $t : \hat{\mathcal{R}} \to \hat{\mathcal{R}}$ be a transformation such that $d^{\mathcal{R}}(R, t(R)) \leq \epsilon$ for all $R \in \hat{\mathcal{R}}$, and let $g = f \circ t$. To show that $f$ is $\epsilon$-robust to misspecification with $g$, we need to show that:

1. For all $R_1, R_2 \in \hat{\mathcal{R}}$, if $f(R_1) = g(R_2)$ then $d^{\mathcal{R}}(R_1, R_2) \leq \epsilon$.

2. For all $R_1, R_2 \in \hat{\mathcal{R}}$, if $f(R_1) = f(R_2)$ then $d^{\mathcal{R}}(R_1, R_2) \leq \epsilon$.

3. For all $R_1 \in \hat{\mathcal{R}}$ there exists an $R_2 \in \hat{\mathcal{R}}$ such that $f(R_2) = g(R_1)$.

4. There exists $R_1, R_2 \in \hat{\mathcal{R}}$ such that $f(R_1) \neq g(R_2)$.

For the first condition, suppose $f(R_1) = g(R_2)$, which implies that $f(R_1) = f \circ t(R_2)$. By assumption, we have that if $f(R) = f(R')$, then $d^{\mathcal{R}}(R, R') = 0$. This implies that $d^{\mathcal{R}}(R_1, t(R_2)) = 0$. Moreover, we have that $d^{\mathcal{R}}(R, t(R)) \leq \epsilon$ for all $R$; this implies that $d^{\mathcal{R}}(R_2, t(R_2)) \leq \epsilon$. By the triangle inequality, we then have that $d^{\mathcal{R}}(R_1, R_2) \leq 0 + \epsilon = \epsilon$. Since $R_1$ and $R_2$ were chosen arbitrarily, this means that condition 1 holds. For condition 2, note that we by assumption have that if $f(R_1) = f(R_2)$, then $d^{\mathcal{R}}(R_1, R_2) = 0$. Since $0 \leq \epsilon$, this implies that condition 2 holds. For condition 3, let $R_1$ be any reward function, and let $R_2 = t(R_1)$. Now $f(R_2) = g(R_1)$. Since $R_1$ was chosen arbitrarily, this means that condition 3 is satisfied. Condition 4 is satisfied by direct assumption. We have thus shown that if $g = f \circ t$ for some $t : \hat{\mathcal{R}} \to \hat{\mathcal{R}}$ such that $d^{\mathcal{R}}(R, t(R)) \leq \epsilon$ for all $R \in \hat{\mathcal{R}}$ and such that $f \neq g$, then $f$ is $\epsilon$-robust to misspecification with $g$ (as defined by $d^{\mathcal{R}}$).

For the other direction, let $f$ be $\epsilon$-robust to misspecification with $g$ (as defined by $d^{\mathcal{R}}$). For each $y \in \text{Im}(g)$, let $R_y \in \hat{\mathcal{R}}$ be some reward function such that $f(R_y) = y$; since $\text{Im}(g) \subseteq \text{Im}(f)$, such an $R_y \in \hat{\mathcal{R}}$ always exists. Now let $t$ be the function that maps each $R \in \hat{\mathcal{R}}$ to $R_{g(R)}$. Since by construction $g(R) = f(R_{g(R)})$, and since $f$ is $\epsilon$-robust to misspecification with $g$ on $\hat{\mathcal{R}}$, we have that $d^{\mathcal{R}}(R, R_{g(R)}) \leq \epsilon$. Since by construction $t(R) = R_{g(R)}$, this means that $d^{\mathcal{R}}(R, t(R)) \leq \epsilon$. Thus $t : \hat{\mathcal{R}} \to \hat{\mathcal{R}}$ satisfies the condition that $d^{\mathcal{R}}(R, t(R)) \leq \epsilon$. Moreover, since $f$ is $\epsilon$-robust to misspecification with $g$, we have that $f \neq g$. Finally, note that $g = f \circ t$. This completes the proof of the other direction, which means that we are done.    □

**Proposition 3.** *A transformation $t : \mathcal{R} \to \mathcal{R}$ satisfies that $d^{\text{STARC}}_{\tau,\gamma}(R, t(R)) \leq \epsilon$ for all $R \in \mathcal{R}$ if and only if $t$ can be expressed as $t_1 \circ \cdots \circ t_{n-1} \circ t_n \circ t_{n+1} \circ \cdots \circ t_m$ for some $n$ and $m$ where*

$$||R - t_n(R)||_2 \leq ||c^{\text{STARC}}_{\tau,\gamma}(R)||_2 \cdot \sin(2 \arcsin(\epsilon/2))$$

*for all $R$, and for all $i \neq n$ and all $R$, we have that $R$ and $t_i(R)$ differ by potential shaping (with $\gamma$), $S'$-redistribution (with $\tau$), or positive linear scaling.*

*Proof.* For the first direction, suppose $d^{\text{STARC}}_{\tau,\gamma}(R, t(R)) \leq \epsilon$ for all $R \in \mathcal{R}$, and let $R$ be an arbitrarily selected reward function. We will show that it is possible to navigate from $R$ to $t(R)$ using the described transformations.

Recall that $d_{\tau,\gamma}^{\mathrm{STARC}}(R_1, R_2)$ is computed by first applying $c_{\tau,\gamma}^{\mathrm{STARC}}$ to both $R_1$ and $R_2$, then normalising the resulting vectors, and finally measuring their $\ell^2$-distance. This means that $s_{\tau,\gamma}^{\mathrm{STARC}}(R)$ and $s_{\tau,\gamma}^{\mathrm{STARC}}(t(R))$ can be placed in the following diagram, where $\epsilon' \le \epsilon$:

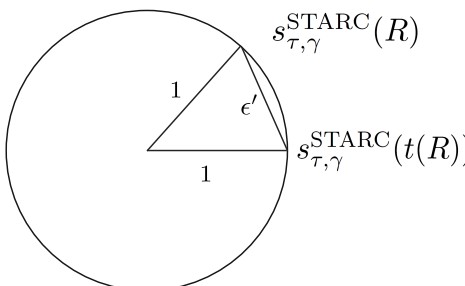

Now, elementary trigonometry tells us that $\theta = 2\arcsin(\epsilon'/2)$. Moreover, suppose we extend $s_{\tau,\gamma}^{\mathrm{STARC}}(R)$ to make the triangle a right triangle, as follows:

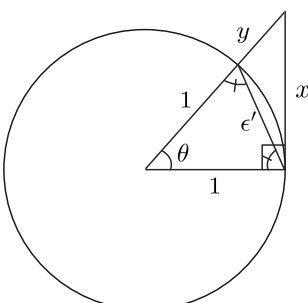

Here elementary trigonometry again tells us that $x/(1 + y) = \sin(2\arcsin(\epsilon'/2))$, or that $x = (1 + y)\sin(2\arcsin(\epsilon'/2))$. This means that we can go from $R$ to $t(R)$ as follows:

1. Apply $c_{\tau,\gamma}^{\mathrm{STARC}}$. Since $R$ and $c_{\tau,\gamma}^{\mathrm{STARC}}(R)$ differ by potential shaping and $S'$-redistribution, this transformation can be expressed as a combination of potential shaping and $S'$-redistribution. Call the resulting vector $R'$.

2. Normalise $R'$, so that its magnitude is 1. This transformation is an instance of positive linear scaling. Call the resulting vector $R''$.

3. Scale $R''$ until it forms a right triangle with $s_{\tau,\gamma}^{\mathrm{STARC}}(t(R))$. This transformation is an instance of positive linear scaling. Call the resulting vector $R'''$.

4. Move from $R'''$ to $s_{\tau,\gamma}^{\text{STARC}}(t(R))$. This will move $R'''$ by $(1+y)\sin(2\arcsin(\epsilon'/2))$, where $(1+y) = ||R'''||_2$. Moreover, since $R'''$ is in the image of $c_{\tau,\gamma}^{\text{STARC}}$, we have that $R''' = c_{\tau,\gamma}^{\text{STARC}}(R''')$, and so $||R'''||_2 = ||c_{\tau,\gamma}^{\text{STARC}}(R''')||_2$. This means that $R'''$ is moved by $||c_{\tau,\gamma}^{\text{STARC}}(R''')||_2 \cdot \sin(2\arcsin(\epsilon'/2))$. Since $\epsilon' \leq \epsilon \leq \pi/2$, this means that $||R''' - s_{\tau,\gamma}^{\text{STARC}}(t(R))||_2 \leq ||c_{\tau,\gamma}^{\text{STARC}}(R''')||_2 \cdot \sin(2\arcsin(\epsilon/2))$.

5. Move from $s_{\tau,\gamma}^{\text{STARC}}(t(R))$ to $c_{\tau,\gamma}^{\text{STARC}}(t(R))$. Since $s_{\tau,\gamma}^{\text{STARC}}(t(R))$ is simply a normalised version of $c_{\tau,\gamma}^{\text{STARC}}(t(R))$, this is an instance of positive linear scaling.

6. Move from $c_{\tau,\gamma}^{\text{STARC}}(t(R))$ to $t(R)$. Since $t(R)$ and $c_{\tau,\gamma}^{\text{STARC}}(t(R))$ differ by potential shaping and $S'$-redistribution, this transformation can be expressed as a combination of potential shaping and $S'$-redistribution.

Thus, for an arbitrary reward function $R$, we can find a series of transformations that fit the given description. This completes the first direction.

For the other direction, suppose $t$ can be expressed as $t_1 \circ \cdots \circ t_{n-1} \circ t_n \circ t_{n+1} \circ \cdots \circ t_m$ where

$$||R - t_n(R)||_2 \leq ||c_{\tau,\gamma}^{\text{STARC}}(R)||_2 \cdot \sin(2\arcsin(\epsilon/2))$$

for all $R$, and for all $i \neq n$ and all $R$, we have that $R$ and $t_i(R)$ differ by potential shaping (with $\gamma$), $S'$-redistribution (with $\tau$), or positive linear scaling.

Recall that $d_{\tau,\gamma}^{\text{STARC}}$ is invariant to potential shaping (with $\gamma$), $S'$-redistribution (with $\tau$), and positive linear scaling; this means that $d_{\tau,\gamma}^{\text{STARC}}(R, t_i(R)) = 0$ for $i \neq n$.

For $t_n$, recall that $c_{\tau,\gamma}^{\text{STARC}}$ is a linear orthogonal projection; this means that $||c_{\tau,\gamma}^{\text{STARC}}(R_1) - c_{\tau,\gamma}^{\text{STARC}}(R_2)||_2 \leq ||R_1 - R_2||_2$. As such, if $||R - t_n(R)||_2 \leq ||c_{\tau,\gamma}^{\text{STARC}}(R)||_2 \cdot \sin(2\arcsin(\epsilon/2))$, then $||c_{\tau,\gamma}^{\text{STARC}}(R) - c_{\tau,\gamma}^{\text{STARC}}(t_n(R))||_2 \leq ||c_{\tau,\gamma}^{\text{STARC}}(R)||_2 \cdot \sin(2\arcsin(\epsilon/2))$ as well. Consider the following diagram:

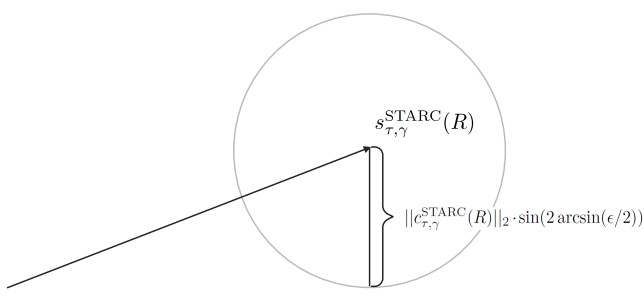

Now $c_{\tau,\gamma}^{\text{STARC}}(t_n(R))$ is located within circle in the diagram above. The vector within this circle that maximises the distance to $c_{\tau,\gamma}^{\text{STARC}}(R)$ *after normalisation* lies on the tangent of the circle:

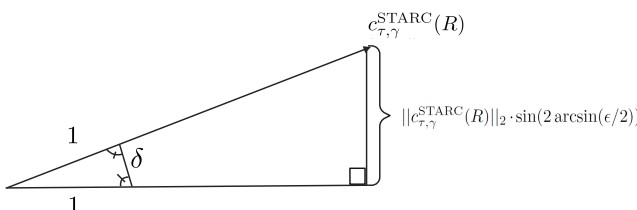

Elementary trigonometry now tells us that

$$\sin(\theta) = \frac{||c_{\tau,\gamma}^{\text{STARC}}(R)||_2 \cdot \sin(2\arcsin(\epsilon/2))}{||c_{\tau,\gamma}^{\text{STARC}}(R)||_2},$$

which gives that $\theta = 2\arcsin(\epsilon/2)$. From this, we have that $\delta = \epsilon$, and so $d_{\tau,\gamma}^{\text{STARC}}(R, t(R)) \leq \epsilon$. This completes the other direction, and hence the proof. □

**Corollary 1.** *Let $\hat{\mathcal{R}}$ be a set of reward functions, $\tau$ be a transition function, $\gamma$ a discount factor, $\beta$ a temperature parameter, and $\alpha$ a weight parameter. Moreover, let $\hat{T}_\epsilon$ be the set of all functions $t : \mathcal{R} \to \mathcal{R}$ that satisfy Proposition 3, and additionally satisfy that $t(R) \in \hat{\mathcal{R}}$ for all $R \in \hat{\mathcal{R}}$. Then $b_{\tau,\gamma,\beta} : \hat{\mathcal{R}} \to \Pi$ is $\epsilon$-robust to misspecification with $g$ (as defined by $d_{\tau,\gamma}^{\text{STARC}}$) if and only if $g = b_{\tau,\gamma,\beta} \circ t$ for some $t \in \hat{T}_\epsilon$, and $c_{\tau,\gamma,\alpha} : \hat{\mathcal{R}} \to \Pi$ is $\epsilon$-robust to misspecification with $g$ (as defined by $d_{\tau,\gamma}^{\text{STARC}}$) if and only if $g = c_{\tau,\gamma,\alpha} \circ t$ for some $t \in \hat{T}_\epsilon$.*

*Proof.* Immediate from Theorem 1 and Proposition 3. □

**Proposition 4.** *Unless $|\mathcal{S}| = 1$ and $|\mathcal{A}| = 2$, then for any $\tau$ and any $\gamma$ there exists an $E > 0$ such that for all $\epsilon < E$, there is no behavioural model $g$ such that $o_{\tau,\gamma}$ is $\epsilon$-robust to misspecification with $g$ (as defined by $d_{\tau,\gamma}^{\text{STARC}}$).*

*Proof.* We will first show that unless $|\mathcal{S}| = 1$ and $|\mathcal{A}| = 2$, there exists reward functions $R_1, R_2$ and an $E > 0$ such that $o_{\tau,\gamma}(R_1) = o_{\tau,\gamma}(R_2)$, but $d_{\tau,\gamma}^{\text{STARC}}(R_1, R_2) = E$. In particular, note that if $|\mathcal{S}| \geq 2$ or $|\mathcal{A}| \geq 3$, then there exists uncountably many reward functions that do not have the same ordering of policies. Moreover, also note that $\text{Im}(o_{\tau,\gamma})$ is finite. By the pigeonhole principle, this means that there must exist a policy $\pi \in \text{Im}(o_{\tau,\gamma})$ and reward functions $R_1, R_2$ such that $o_{\tau,\gamma}(R_1) = o_{\tau,\gamma}(R_2) = \pi$, and such that $R_1$ and $R_2$ do not have the same ordering of policies. Moreover, recall that $d_{\tau,\gamma}^{\text{STARC}}(R_1, R_2) = 0$ if and only if $R_1$ and $R_2$ have the same ordering of policies. Thus, unless $|\mathcal{S}| = 1$ and $|\mathcal{A}| = 2$, there exists reward functions $R_1, R_2$ such that $o_{\tau,\gamma}(R_1) = o_{\tau,\gamma}(R_2)$, but $d_{\tau,\gamma}^{\text{STARC}}(R_1, R_2) = E > 0$. Thus $o_{\tau,\gamma}$ violates condition 2 of Definition 1 for all $\epsilon < E$. □

### F.2 PERTURBATION ROBUSTNESS

**Theorem 2.** *Let $\hat{\mathcal{R}}$ be a set of reward functions, let $f : \hat{\mathcal{R}} \to \Pi$ be a behavioural model, let $d^{\mathcal{R}}$ be a pseudometric on $\hat{\mathcal{R}}$, and let $d^\Pi$ be a pseudometric on $\Pi$. Then $f$ is $\epsilon$-robust to $\delta$-perturbation (as defined by $d^{\mathcal{R}}$ and $d^\Pi$) if and only if $f$ is $\epsilon/\delta$-separating (as defined by $d^{\mathcal{R}}$ and $d^\Pi$).*

*Proof.* For the first direction, suppose $f$ is $\epsilon/\delta$-separating, and let $g$ be a $\delta$-perturbation of $f$ with $\text{Im}(g) \subseteq \text{Im}(f)$. We will show that $f$ and $g$ satisfy the conditions of Definition 1. For the first

condition, let $R_1, R_2$ be two arbitrary reward functions in $\hat{\mathcal{R}}$ such that $f(R_1) = g(R_2)$. Since $g$ is a $\delta$-perturbation of $f$, we have that $d^\Pi(g(R_2), f(R_2)) \leq \delta$. Since $f(R_1) = g(R_2)$, straightforward substitution thus gives us that $d^\Pi(f(R_1), f(R_2)) \leq \delta$. Since $f$ is $\epsilon/\delta$-separating, this means that $d^\mathcal{R}(R_1, R_2) \leq \epsilon$. Since $R_1$ and $R_2$ were chosen arbitrarily, this means that if $f(R_1) = g(R_2)$ then $d^\mathcal{R}(R_1, R_2) \leq \epsilon$. Thus, the first condition of Definition 1 holds. For the second condition, note that if $f(R_1) = f(R_2)$, then $d^\Pi(f(R_1), f(R_2)) = 0 \leq \delta$. Since $f$ is $\epsilon/\delta$-separating, this means that $d^\mathcal{R}(R_1, R_2) \leq \epsilon$, which means that the second condition is satisfied as well. The third condition is satisfied, since we assume that $\mathrm{Im}(g) \subseteq \mathrm{Im}(f)$, and the fourth condition is satisfied by the definition of $\delta$-perturbations. This means that $f$ and $g$ satisfy all the conditions of Definition 1, and thus $f$ is $\epsilon$-robust to misspecification with $g$. Since $g$ was chosen arbitrarily, this means that $f$ is $\epsilon$-robust to misspecification with any $\delta$-perturbation $g$ such that $\mathrm{Im}(g) \subseteq \mathrm{Im}(f)$. Thus $f$ is $\epsilon$-robust to $\delta$-perturbation.

For the second direction, suppose $f$ is *not* $\epsilon/\delta$-separating. This means that there exist $R_1, R_2 \in \hat{\mathcal{R}}$ such that $d^\mathcal{R}(R_1, R_2) > \epsilon$ and $d^\Pi(f(R_1), f(R_2)) \leq \delta$. Now let $g : \hat{\mathcal{R}} \to \hat{\mathcal{R}}$ be the behavioural model where $g(R_1) = f(R_2)$, $g(R_2) = f(R_1)$, and $g(R) = f(R)$ for all $R \notin \{R_1, R_2\}$. Now $g$ is a $\delta$-perturbation of $f$. However, $f$ is not $\epsilon$-robust to misspecification with $g$, since $g(R_1) = f(R_2)$, but $d^\mathcal{R}(R_1, R_2) > \epsilon$. Thus, if $f$ is not $\epsilon/\delta$-separating then $f$ is not $\epsilon$-robust to $\delta$-perturbation, which in turn means that if $f$ is $\epsilon$-robust to $\delta$-perturbation, then $f$ is must be $\epsilon/\delta$-separating. $\qquad\square$

**Theorem 3.** *Let $d^\mathcal{R}$ be $d_{\tau,\gamma}^{\mathrm{STARC}}$, and let $d^\Pi$ be a pseudometric on $\Pi$ which satisfies the condition that for all $\delta_1$ there exists a $\delta_2$ such that if $||\pi_1 - \pi_2||_2 < \delta_2$ then $d^\Pi(\pi_1, \pi_2) < \delta_1$. Let $c$ be any positive constant, and let $\hat{\mathcal{R}}$ be a set of reward functions such that if $||R||_2 = c$ then $R \in \hat{\mathcal{R}}$. Let $f : \hat{\mathcal{R}} \to \Pi$ be continuous. Then $f$ is not $\epsilon/\delta$-separating for any $\epsilon < 1$ or $\delta > 0$.*

*Proof.* Let $R$ be a non-trivial reward function that is orthogonal to all trivial reward functions. Since the set of all trivial reward functions form a linear subspace, such a reward function $R$ exists. Note that $R$ must not necessarily be contained in $\hat{\mathcal{R}}$.

We now have that for any positive constant $\epsilon$, it is the case that $\epsilon R$ and $-\epsilon R$ have the opposite ordering of policies, and thus $d_{\tau,\gamma}^{\mathrm{STARC}}(\epsilon R, -\epsilon R) = 1$. Next, let $R_\Phi$ be some potential-shaping reward function such that $||\epsilon R + R_\Phi||_2 = c$. Since potential shaping does not change the ordering of policies, we have that $\epsilon R + R_\Phi$ and $\epsilon R + R_\Phi$ must have the opposite ordering of policies as well, and so $d_{\tau,\gamma}^{\mathrm{STARC}}(\epsilon R + R_\Phi, -\epsilon R + R_\Phi) = 1$. Moreover, since $R_\Phi$ is trivial, we have that both $\epsilon R$ and $-\epsilon R$ are orthogonal to $R_\Phi$, and so $||-\epsilon R + R_\Phi||_2 = c$ as well. Since $||\epsilon R + R_\Phi||_2 = ||\epsilon R + R_\Phi||_2 = c$, we have that both $\epsilon R + R_\Phi$ and $-\epsilon R + R_\Phi$ are contained in $\hat{\mathcal{R}}$.

Let $\delta_1$ be any positive constant. By assumption, there exists a $\delta_2$ such that if $||\pi_1 - \pi_2||_2 < \delta_2$ then $d^\Pi(\pi_1, \pi_2) < \delta_1$. Moreover, since $f$ is continuous, there exists an $\epsilon_1$ such that if $||R_1 - R_2||_2 < \epsilon_1$, then $||f(R_1) - f(R_2)||_2 < \delta_2$. Next, note that by making $\epsilon$ sufficiently small, we can ensure that the $\ell^2$-distance between $\epsilon R + R_\Phi$ and $\epsilon R + R_\Phi$ is arbitrarily small (and, in particular, less than $\epsilon_1$).

Thus, for any positive $\delta$ there exist reward functions $\epsilon R + R_\Phi$ and $-\epsilon R + R_\Phi$ that are both contained in $\hat{\mathcal{R}}$, such that $d^\Pi(f(\epsilon R + R_\Phi), f(-\epsilon R + R_\Phi)) < \delta$, and such that $d_{\tau,\gamma}^{\mathrm{STARC}}(\epsilon R + R_\Phi, -\epsilon R + R_\Phi) = 1$. Thus $f$ is not $\epsilon/\delta$-separating for any $\delta > 0$ and any $\epsilon < 1$. $\qquad\square$

### F.3 MISSPECIFIED PARAMETERS

**Lemma 1.** *Let $\hat{\mathcal{R}}$ be a set of reward functions, let $f, g : \hat{\mathcal{R}} \to \Pi$ be two behavioural models, and let $d^\mathcal{R}$ be a pseudometric on $\hat{\mathcal{R}}$. Suppose $f$ is $\epsilon$-robust to misspecification with $g$ (as defined by $d^\mathcal{R}$). Then if $g(R_1) = g(R_2)$, we have that $d^\mathcal{R}(R_1, R_2) \leq 2\epsilon$.*

*Proof.* Let $f, g : \hat{\mathcal{R}} \to \hat{\mathcal{R}}$ be two behavioural models, and suppose $f$ is $\epsilon$-robust to misspecification with $g$. Let $R_1, R_2 \in \hat{\mathcal{R}}$ be any two reward functions such that $g(R_1) = g(R_2)$. From condition 3 in Definition 1, we have that there must be a reward function $R_3$ such that $f(R_3) = g(R_1) = g(R_2)$. From condition 1 in Definition 1, we have that $d^\mathcal{R}(R_3, R_1) \leq \epsilon$ and $d^\mathcal{R}(R_3, R_2) \leq \epsilon$. The triangle inequality then implies that $d^\mathcal{R}(R_1, R_2) \leq 2\epsilon$. $\qquad\square$

**Lemma 2.** *If $f_\gamma : \mathcal{R} \to \Pi$ is invariant to potential shaping with $\gamma$, then for all $\tau$ and all $\gamma_1, \gamma_2$ such that $\gamma_1 \neq \gamma_2$ and $\tau$ is non-trivial, then there exists a reward function $R^\dagger$ such that $f_{\gamma_1}(R) = f_{\gamma_1}(R + \alpha R^\dagger)$ for all $R \in \mathcal{R}$ and all $\alpha \in \mathbb{R}$.*

*Proof.* Analogous to the proof of Lemma A.18 in Skalse & Abate (2023). $\square$

**Lemma 3.** *If $f_\tau : \mathcal{R} \to \Pi$ is invariant to $S'$-redistribution with $\tau$, then for all $\gamma$ and all $\tau_1, \tau_2$ such that $\tau_1 \neq \tau_2$, there exists a reward function $R^\dagger$ that is non-trivial under $\tau_2$ and $\gamma$, such that $f_{\tau_1}(R) = f_{\tau_1}(R + \alpha R^\dagger)$ for all $R \in \mathcal{R}$ and all $\alpha \in \mathbb{R}$.*

*Proof.* Since $f_{\tau_1}$ is invariant to $S'$-redistribution with $\tau_1$, we have that $f_{\tau_1}(R_1) = f_{\tau_1}(R_2)$ for any two reward functions $R_1, R_2$ such that

$$\mathbb{E}_{S' \sim \tau_1(s,a)}[R_1(s, a, S')] = \mathbb{E}_{S' \sim \tau_1(s,a)}[R_2(s, a, S')].$$

Note that $R_1$ and $R_2$ satisfy this condition if and only if

$$\mathbb{E}_{S' \sim \tau_1(s,a)}[(R_2 - R_1)(s, a, S')] = 0.$$

That is to say, if $R'$ is a reward function such that $\mathbb{E}_{S' \sim \tau_1(s,a)}[R'(s, a, S')] = 0$, then $f_{\tau_1}(R) = f_{\tau_1}(R + R')$ for all $R$. Next, note that the set of all such reward functions $R'$ form a linear subspace of $\mathcal{R}$, with $|\mathcal{S}||\mathcal{A}|(|\mathcal{S}| - 1)$ dimensions. We will show that this subspace contains reward functions that are non-trivial under $\gamma$ and $\tau_2$.

Since $\tau_1 \neq \tau_2$, we have that there exists a state $s$ and action $a$ such that $\tau_1(s, a) \neq \tau_2(s, a)$. Let $R^\dagger$ be a reward function that is $0$ everywhere, except that $\mathbb{E}_{S' \sim \tau_1(s,a)}[R'(s, a, S')] = 0$, and $\mathbb{E}_{S' \sim \tau_2(s,a)}[R'(s, a, S')] = 1$. Note that there is always a solution to this system of linear equations. In particular, the values of $R^\dagger(s, a, s')$ for each transition $s, a, s'$ form a $|\mathcal{S}|$-dimensional vector space. The set of all values for these variables that satisfy $\mathbb{E}_{S' \sim \tau_1(s,a)}[R'(s, a, S')] = 0$ form an $(|\mathcal{S}| - 1)$-dimensional linear subspace, and the set of all values that satisfy $\mathbb{E}_{S' \sim \tau_2(s,a)}[R'(s, a, S')] = 1$ form an $(|\mathcal{S}| - 1)$-dimensional affine subspace. These two sets must intersect, unless they are parallel. However, since $\sum_{s'} \mathbb{P}(\tau_1(s, a) = s') = \sum_{s'} \mathbb{P}(\tau_2(s, a) = s') = 1$, they cannot be parallel. Thus, such a reward function $R^\dagger$ must exist.

It is clear that $R^\dagger$ is non-trivial under $\gamma$ and $\tau_2$. To spell it out; since all states are reachable under $\tau_2$ and $\mu_0$, there exists a policy $\pi$ that visits state $s$ with positive probability. Let $\pi_1$ and $\pi_2$ be two policies that are identical to $\pi$ everywhere, except that $\pi_1$ takes action $a$ with probability $1$ in state $s$, and $\pi_2$ takes action $a$ with probability $0$ in state $s$. Then $J^\dagger(\pi_1) > J^\dagger(\pi_2)$. Moreover, since $\mathbb{E}_{S' \sim \tau_1(s,a)}[R^\dagger(s, a, S')] = 0$ for all $s$ and $a$, we have that $R$ and $R + \alpha R^\dagger$ differ by $S'$-redistribution (with $\tau_1$) for all reward functions $R \in \mathcal{R}$ and all scalars $\alpha \in \mathbb{R}$. Thus $f_{\tau_1}(R) = f_{\tau_1}(R + \alpha R^\dagger)$.

Thus, if $f_\tau : \mathcal{R} \to \Pi$ is invariant to $S'$-redistribution with $\tau$, then for all $\gamma$ and all $\tau_1, \tau_2$ such that $\tau_1 \neq \tau_2$, there exists a reward function $R^\dagger$ that is non-trivial under $\tau_2$ and $\gamma$, such that $f_{\tau_1}(R) = f_{\tau_1}(R + \alpha R^\dagger)$ for all $R \in \mathcal{R}$ and all $\alpha \in \mathbb{R}$. $\square$

**Theorem 4.** *If $f_\gamma : \mathcal{R} \to \Pi$ is invariant to potential shaping with $\gamma$, and $\gamma_1 \neq \gamma_2$, then $f_{\gamma_1}$ is not $\epsilon$-robust to misspecification with $f_{\gamma_2}$ under $d^{\text{STARC}}_{\tau, \gamma_3}$ for any non-trivial $\tau$, any $\gamma_3$, and any $\epsilon < 0.5$.*

*Proof.* If $\gamma_1 \neq \gamma_2$, then either $\gamma_1 \neq \gamma_3$ or $\gamma_2 \neq \gamma_3$.

If $\gamma_1 \neq \gamma_3$, then Lemma 2 implies that there exists a reward function $R^\dagger$ that is non-trivial under $\tau$ and $\gamma_3$, such that $f_{\gamma_1}(R) = f_{\gamma_1}(R + \alpha R^\dagger)$ for all $R \in \mathcal{R}$ and all $\alpha \in \mathbb{R}$. This means that $f_{\gamma_1}(R^\dagger) = f_{\gamma_1}(-R^\dagger)$ and $d^{\text{STARC}}_{\tau, \gamma_3}(R^\dagger, -R^\dagger) = 1$. Thus $f_{\gamma_1}$ violates condition 2 of Definition 1 for all $\epsilon < 1$.

If $\gamma_2 \neq \gamma_3$, then Lemma 2 implies that there exists a reward function $R^\dagger$ that is non-trivial under $\tau$ and $\gamma_3$, such that $f_{\gamma_2}(R) = f_{\gamma_2}(R + \alpha R^\dagger)$ for all $R \in \mathcal{R}$ and all $\alpha \in \mathbb{R}$. This means that $f_{\gamma_2}(R^\dagger) = f_{\gamma_2}(-R^\dagger)$ and $d^{\text{STARC}}_{\tau, \gamma_3}(R^\dagger, -R^\dagger) = 1$. Then Lemma 1 implies that there can be no $f$ that is $\epsilon$-robust to misspecification with $f_{\gamma_2}$ (as defined by $d^{\text{STARC}}_{\tau, \gamma_3}$) for any $\epsilon < 0.5$. $\square$

**Theorem 5.** *If $f_\tau : \mathcal{R} \to \Pi$ is invariant to $S'$-redistribution with $\tau$, and $\tau_1 \neq \tau_2$, then $f_{\tau_1}$ is not $\epsilon$-robust to misspecification with $f_{\tau_2}$ under $d^{\text{STARC}}_{\tau_3, \gamma}$ for any $\tau_3$, any $\gamma$, and any $\epsilon < 0.5$.*

*Proof.* If $\tau_1 \neq \tau_2$, then either $\tau_1 \neq \tau_3$ or $\tau_2 \neq \tau_3$.

If $\tau_1 \neq \tau_3$, then Lemma 3 implies that there exists a reward function $R^\dagger$ that is non-trivial under $\tau_3$ and $\gamma$, such that $f_{\tau_1}(R) = f_{\tau_1}(R + \alpha R^\dagger)$ for all $R \in \mathcal{R}$ and all $\alpha \in \mathbb{R}$. This means that $f_{\tau_1}(R^\dagger) = f_{\tau_1}(-R^\dagger)$ and $d_{\tau_3,\gamma}^{\mathrm{STARC}}(R^\dagger, -R^\dagger) = 1$. Thus $f_{\tau_1}$ violates condition 2 of Definition 1 for all $\epsilon < 1$.

If $\tau_2 \neq \tau_3$, then Lemma 3 implies that there exists a reward function $R^\dagger$ that is non-trivial under $\tau_3$ and $\gamma$, such that $f_{\tau_2}(R) = f_{\tau_2}(R + \alpha R^\dagger)$ for all $R \in \mathcal{R}$ and all $\alpha \in \mathbb{R}$. This means that $f_{\tau_2}(R^\dagger) = f_{\tau_2}(-R^\dagger)$ and $d_{\tau_3,\gamma}^{\mathrm{STARC}}(R^\dagger, -R^\dagger) = 1$. Then Lemma 1 implies that there can be no $f$ that is $\epsilon$-robust to misspecification with $f_{\tau_2}$ (as defined by $d_{\tau_3,\gamma}^{\mathrm{STARC}}$) for any $\epsilon < 0.5$. $\square$

## G CONNECTING OUR ANALYSIS TO EARLIER PROPOSALS

In this section, we will explain how to connect the results of Skalse & Abate (2023) to our results in a rigorous way. Skalse & Abate (2023) assume that we have a partition $P$ on $\mathcal{R}$, which of course corresponds to an equivalence relation $\equiv_P$, and say that two reward functions $R_1, R_2$ should be considered to be "close" if $R_1 \equiv_P R_2$. Like us, they consider functions $f, g : \mathcal{R} \to \Pi$ that take a reward function and return a policy. They then say that $f$ is "$P$-robust to misspecification with $g$" if each of the following conditions hold:

1. $f(R_1) = g(R_2) \implies R_1 \equiv_P R_2$.
2. $f(R_1) = f(R_2) \implies R_1 \equiv_P R_2$.
3. For all $R_1$ there exists an $R_2$ such that $f(R_2) = g(R_1)$.
4. $f \neq g$.

Note that this definition is analogous to Definition 1, except that an equivalence relation $P$ plays the role that a pseudometric $d^\mathcal{R}$ does in our framework. Next, note that we for any pseudometric $d^\mathcal{R}$ can define an equivalence relation $\equiv_P$ such that $R_1 \equiv_P R_2$ if and only if $d^\mathcal{R}(R_1, R_2) = 0$. In that case, we would have that $f$ is $P$-robust to misspecification with $g$ (in the terminology of Skalse & Abate, 2023) if and only if $f$ is 0-robust to misspecification with $g$ (as evaluated by $d^\mathcal{R}$) in our terminology (i.e. Definition 1). Moreover, if $f$ is 0-robust to misspecification with $g$, then it of course follows that $f$ is $\epsilon$-robust to misspecification with $g$ for all $\epsilon \geq 0$. In this way, their results can be expressed within our more general framework.

Next, also recall that $d_{\tau,\gamma}^{\mathrm{STARC}}(R_1, R_2) = 0$ if and only if $R_1$ and $R_2$ induce the same ordering of policies (under $\tau$ and $\gamma$). Skalse & Abate (2023) use "$\mathrm{ORD}^\mathcal{M}$" to denote this equivalence relation. Thus, if $f$ is $\mathrm{ORD}^\mathcal{M}$-robust to misspecification with $g$ (as defined by Skalse & Abate, 2023) then $f$ is $\epsilon$-robust to misspecification with $g$ (as evaluated by $d_{\tau,\gamma}^{\mathrm{STARC}}$) for all $\epsilon \geq 0$.

