# OpenReview forum: "Quantifying the Sensitivity of Inverse Reinforcement Learning to Misspecification"
_ICLR.cc/2024/Conference — ICLR 2024 poster_

### Official Review · Reviewer_CDtC · 2023-10-28

**Soundness:** 3 good
**Presentation:** 3 good
**Contribution:** 3 good
**Rating:** 6
**Confidence:** 2

**Summary:**

This paper presents a theoretical framework aimed at investigating the discrepancy or 'misspecification' between the assumed and actual behavioural models during the Inverse Reinforcement Learning (IRL) process. Moreover, it explores how this misspecification impacts the reward obtained through the IRL algorithm.

This research primarily concentrates on the most prevalent behavioural models, namely optimality, Boltzmann-rationality, and causal entropy maximisation.

In the initial segment of the theoretical framework, it introduces a formal definition of 'misspecification robustness' and the 'STARC-metric'. This metric is used to measure the differences between a reward function that remains invariant to potential shaping, redistribution, and possibly rescaling. Leveraging these definitions, the paper outlines the necessary and sufficient conditions for identifying the degree of robustness against misspecification between two behavioural models.

The second segment of the theoretical framework presumes a distance between the returns of behavioural models (policies). It examines the capability of the IRL algorithm to learn precise rewards under two conditions: 1) when the behavioural models are misspecified, and 2) when there is a gap in their returns. The findings suggest that no continuous behavioural model can maintain robustness against these two levels of misspecification.

In the final part of the theoretical framework, the paper delves into the scenarios where the parameters of behavioural models are misspecified. The research primarily focuses on the discount parameter, demonstrating that behavioural models are, in principle, highly sensitive to even minor misspecifications of two of their core parameters.

**Strengths:**

1. The issue of misspecification between the assumed and actual behavioural models in IRL reresents a significant and underexplored challenge, to the best of the reviewer's knowledge.

2. The paper is logically structured and easily navigable, with definitions clearly articulated and supplemented by explanations. These help the reader to intuitively comprehend the main theses and their implications in terms of characterizing the robustness of misspecification in the IRL problem.

3. The theoretical framework is presented with clarity, and the main claims appear generally sound.

**Weaknesses:**

1. The inclusion of concrete examples of the transformation would be beneficial. Currently, it is challenging to image the types of transformations that would satisfy Proposition 3, and more importantly, how to interpret these transformations requires clarification.

2. As per the reviewer's understanding, an MCE (Maximum Causal Entropy) policy corresponds to a Boltzmann-rational policy. Specifically, the optimal policy representation in soft-Q learning complies with Boltzmann-rationality. This point may warrant further verification.

3. It would be beneficial to present some empirical results, which could illuminate the consequences of failing to achieve robustness under the misspecification of behavioural models.

**Questions:**

1. In proposition 3, what's the meaning of $\|R,t_n(R)\|_2$? Should it be |R-t_n(R)\|_2?

2. According to the results of this paper, all the commonly applied behavior models are not $\epsilon/\delta$-separating, are there any behavior model is $\epsilon/\delta$, maybe under the mild relaxation on the distance under the reward and policy space?

---

> ### Author Response · Authors · 2023-11-12
> **Response**
>
> We thank reviewer CDtC for their review. Our responses to your questions are as follows:
>
> Weaknesses:
> 1. We appreciate that the transformations described in Proposition 3 are quite difficult to visualise. The proof of Proposition 3 includes a geometric description of these transformations, but they are still not easy to understand intuitively. To some extent, this is just a consequece of the mathematics --- we wanted to derive necessary and sufficient conditions, and in this case, those conditions turned out to be somewhat messy. However, we can certainly include a few examples to make Proposition 3 more intuitive. First of all, Proposition 3 is satisfied by any transformation that can be expressed as a combination of potential shaping, positive linear scaling, and S'-redistribution, and these transformations are reasonably easy to understand (see Section 1.2). More generally, a transformation satisfies Proposition 3 if it never changes the policy order of the reward function by a large amount. For example, suppose we have a Bandit MDP with three actions, and that the reward obtained from these actions is $\langle 1, 1-\epsilon, 0\rangle$. A reward transformation that satisfies Proposition 3 might change this reward to $\langle 1-\epsilon, 1, 0\rangle$, for example. We can include a few examples like this, to help make Proposition 3 more intelligible.
> 2. The MCE policy and the Boltzmann-rational policy are not identical, though they are similar in some ways. The Boltzmann-rational policy is given by softmaxing the optimal Q-function, whereas the MCE policy is given by softmaxing the soft Q-function. The optimal Q-function and the soft Q-function are in turn not identical, which means that these two behavioural models are slightly different.
> 3. We agree that more extensive empirical investigation would be interesting, but we consider this to be out of scope for this paper. It would be redundant to verify our positive results (i.e. Theorem 1 and 2) experimentally, as our proofs already tell us what the outcome of such experiments would be. It would be more interesting to verify our negative results (i.e. Theorem 3, 4, and 5). However, the outcome of these experiments would be heavily dependent on the choice of prior probability distribution over the true reward function $R^\star$, and on the choice of inductive bias of the learning algorithm --- our proofs tell us that there for any inductive bias is a prior probability distribution over $R^\star$ under which our bounds are attained with high probability. For Theorem 4 and 5, the converse also holds. Nonetheless, more extensive empirical investigation would certainly be interesting, but we consider this to be out of scope for this paper.
>
>
> Questions:
> 1. Yes, it should be $||R - t_n(R)||_2$. In this paper, we have used the convention where $||x, y||_2$ denotes the metric induced by the $\ell_2$-norm, ie $||x - y||_2$. Since several reviewers found this confusing, we will replace it with the more typical notation.
> 2. The result that all the common behavioural models fail to be $\epsilon/\delta$-separating does depend on the choice of metric(s). Specifically, it is a consequence of the fact that two reward functions can have a small $\ell_2$-norm, even though they have a large STARC-distance (see Appendix D). As such, the proof would not go through if we measure the distance between rewards using the $\ell_2$-norm. However, if we do this, then we would not obtain informative regret bounds (as discussed in Section 2.2). There may be other metrics under which the standard behavioural models are $\epsilon/\delta$-separating, and under which some form of regret bounds or other theoretical guarantees can be obtained. However, no such metric has been proposed in the existing literature. We believe that further exploration of this question is an interesting topic for further work.
>
> We hope that this clarifies our results, and that the reviewer might consider increase their score!

---

> > ### Comment · Reviewer_CDtC · 2023-11-19
> >
> > Thank you for your response; it addresses most of my concerns. I do have a suggestion for the authors. Given your primary interest in the misspecification of behavioral models, it might be beneficial to reflect this more clearly in your title. In the realm of IRL, the term 'misspecification' can imply various things, including, for instance, the misspecification of training and testing dynamics.

---

> > > ### Author Response · Authors · 2023-11-19
> > >
> > > Thank you for this. We can change the title of our paper to "Quantifying the Sensitivity of Inverse Reinforcement Learning to Misspecification of the Behavioural Model". However, note that we also consider misspecification of the environment dynamics in Section 3.3, and our results from Section 3.2 may be informative for when considering other forms of misspecification as well.

---

### Official Review · Reviewer_jn4g · 2023-10-31

**Soundness:** 3 good
**Presentation:** 3 good
**Contribution:** 3 good
**Rating:** 6
**Confidence:** 3

**Summary:**

Inverse reinforcement learning aims to infer a reward function $R$ from a policy $\pi$. To achieve this, IRL needs to make an assumption on the underlying behavior model, which maps reward functions to policies and characterizes the relation between $R$ and $\pi$. However, the behavior model could be misspecified and differs from the true one. This paper is to contribute towards a theoretical understanding on the robustness of IRL to misspecification. First, this paper characterizes the sufficient and necessary condition to ensure that IRL is robust to a general misspecification. Then this paper considers a specific type of misspecification: the behavior model is a perturbation of the true one, and provides the corresponding sufficient and necessary condition. Finally, this paper considers another type of misspecification on the parameters of behavior models and provides a negative result that IRL is often not robust to such misspecification.

**Strengths:**

1. This paper indeed makes notable progress toward theoretical understandings of misspecification in IRL. This paper conducts a systematic analysis and provides complete answers to this problem. The main results are insightful and clean and can help deepen the understanding of IRL.
2. The paper is well-written and easy to follow. Although this paper is a theoretical paper, it provides many intuitive examples and explanations that can help readers easily understand the main results.

**Weaknesses:**

1. In this paper, the authors consider a general IRL procedure that outputs an arbitrary reward function $R$ from the class $\tilde{\mathcal{R}} = \\{R_1: f (R_1) = \pi = g(R_2) \\}$. However, in practice, we often run a specific IRL algorithm such as MaxEnt IRL [Ziebart, 2010], which performs reward selection and outputs the chosen reward. Therefore, it could be more meaningful to analyze the robustness of a specific IRL algorithm to misspecification.

**Questions:**

Typos:
1. In Proposition 3, $\\| R, t_n (R) \\|_2$ should be $\\| R - t_n (R) \\|_2$?

---

> ### Author Response · Authors · 2023-11-12
> **Response**
>
> We would like to thank reviewer jn4g for their review and feedback!
>
> We agree that it could be interesting to provide a more detailed analysis of specific IRL algorithms, such as MaxEnt IRL. However, note that all of our results apply to any algorithm that is based on the MCE model, including MaxEnt IRL. In this sense, our analysis is simply more general than an analysis which is restricted to a particular algorithm.
>
> Of course, in general, the inductive bias of an algorithm can be important for understanding its behaviour --- this is discussed in some more detail in Appendix A.4. However, note that Proposition 1 and 2 together imply that if two reward functions have the same MCE policy, then their STARC distance is 0. This means that if an algorithm is based on the MCE model, then its inductive bias will not affect the quality of the inferred reward function in the limit of infinite data. As such, we would not be able to draw stronger conclusions by incorporating more details about the MaxEnt IRL algorithm.
>
> Yes, in Proposition 3, it should be $||R - t_n(R)||_2$. In this paper, we have used the convention where $||x, y||_2$ denotes the metric induced by the $\ell_2$-norm, ie $||x - y||_2$. Since several reviewers found this confusing, we will replace it with the more typical notation.
>
> We hope that this clarifies our results, and that the reviewer might consider increasing their score.

---

> > ### Comment · Reviewer_jn4g · 2023-11-22
> >
> > Thank the authors for their detailed explanation. After reading the reviews from other reviewers and the responses from the authors, I'd like to keep my current score unchanged.

---

### Official Review · Reviewer_Bsty · 2023-11-01

**Soundness:** 2 fair
**Presentation:** 2 fair
**Contribution:** 2 fair
**Rating:** 6
**Confidence:** 2

**Summary:**

The authors analyze the sensitivity of inverse reinforcement learning, i.e., learning the reward model to misspecification of the behavior model adopted by the agent, i.e., the objective function that the agent is trying to optimize. The authors analyse robustness of IRL to misspecification in the behavior model by first quantifying the impact of misspecification of behavior model in terms of a pseudometric defined on the space of reward functions. Their first theorem states necessary and sufficient conditions for a behavior model to be $\epsilon$ robust to a defined family of behavior models. The authors then state the necessary and sufficient conditions under which a behavioral model is robust to perturbations as their second theorem. Their final theorems state conditions under which behavioral models with perturbations to their parameters are not robust in terms of learning a reward function that preserves preferences.

**Strengths:**

The authors present interesting propositions that will help characterise the robustness of IRL algorithms towards learning preferences.

**Weaknesses:**

1. I am not sure about the extent of novelty in this contribution. The problem of robustness of IRL to behavior model misspecification has been studied in the past as mentioned in the literature survey in the paper. The new pseudometric used by the authors has also been introduced in an earlier paper cited by the authors. If this is correct, and the analysis is the main novel contribution, then I would expect a more detailed comparison of the analysis with the other works in literature.
2. The main contribution of this paper is its theoretical results. The authors hhave stated the theorems, propositions and corollaries in the main paper without any proofs or even proof sketches. Furthermore, there are no numerical experiments conducted to demonstrate the impact of the theoretical findings.

**Questions:**

1. Can't the difference in observed versus predicted behavior of a human assuming a particular behavior model be explained by incorrect or incomplete optimization rather than incorrectness of the behavior model?
2. Doesn't the IRL method also need access to the environment to collect samples?
3. For a given dataset, can't two different behavioral models yield the same reward function? If so, isn't accuracy of reward prediction an insufficient criterion for estimating the validity of the behavioral model? In this paper, do the authors wish to learn a true reward function, albeit woith an incorrect behavior model, or do they wish to establish correctness of the behavior model as well?
4. Are the brackets correctly places in the definition of S' redistribution in the third paragraph of Sec 1.2? If so, I do not understand this definition. Can the authors please clarify?
5. In Point 3 of Defintion 1, should $R_1$ and $R_2$ be necessarily different? If so, why? What happens in the case when $f = g$?
6. Is the orderting of policies mentioned in Proposition 1 a partial ordering?
7. I did not understand the 2-norm of the first term in the equation written in Proposition 3. Can the authors please explain this?
8. What happens to IRL robustness when the true behavior model is Cumulative Prospect Theory and the assumed one is Boltzmann-rationality?

---

> ### Author Response · Authors · 2023-11-12
> **Response**
>
> First of all, we would like to thank reviewer Bsty for your thoughts and feedback! Our responses to your concerns and questions are as follows:
>
> Weaknesses:
> 1. The novelty of our contribution does indeed lie in our analysis (i.e., the results we present in Section 3). While misspecification in IRL has been studied by a handful of recent papers, our analysis is more comprehensive, and is done in a richer formal framework, which lets us infer many novel insights. We have attempted to explain precisely how our work extends earlier work in Section 1.1 and Appendix G, but we can certainly make these sections more detailed. To make this more clear, we have included a more detailed comparison between our paper and earlier work in a separate comment.
>
> 2. All proofs are provided in Section F of the appendix, on pages 20-26. We also provide a high-level intuitive explanation of some of these proofs in Section D and E of the appendix, on pages 17-19. There would not be enough space to fit this in the main part of the text. However, note that it is very common to put long proofs in the appendix. As for experiments, we note that this is a theoretical paper, rather than an empirical paper. It would be redundant to verify our positive results (i.e. Theorem 1 and 2) experimentally, as our proofs already tell us what the outcome of such experiments would be. It would be more interesting to verify our negative results (i.e. Theorem 3, 4, and 5). However, the outcome of these experiments would be heavily dependent on the choice of prior probability distribution over the true reward function $R^\star$, and on the choice of inductive bias of the learning algorithm --- our proofs tell us that there for any inductive bias is a prior probability distribution over $R^\star$ under which our bounds are attained with high probability. For Theorem 4 and 5, the converse also holds. Nonetheless, more extensive empirical investigation would certainly be interesting, but we consider this to be out of scope for this paper.
>
>
> Questions.
> 1. Yes, there are many potential sources of error in the IRL problem. Our work focuses on misspecification of the behavioural model, as we believe this issue to be particularly important and difficult to overcome. However, IRL certainly has many other difficulties as well.
> 2. Yes. This is implicit in our formalism, since the behavioural model depends on the transition function. Treating this dependence as being part of the behavioural model makes it easier to reason about the case where the transition dynamics are themselves misspecified.
> 3. Yes, two behavioural models can produce the same reward function(s) for a given data distribution. However, the aim of our work is not to determine the vailidity of the behavioural model, but rather, to quantify how sensitive IRL is to misspecification of the behavioural model. In other words, we want to know when it is possible to infer a reward function that is "close enough" to the true reward function using a behavioural model that is misspecified.
> 4. The brackets are indeed incorrect, thank you for spotting this. The correct statement should read $E[R_1(s,a,S')] = E[R_2(s,a,S')]$.
> 5. $R_1$ and $R_2$ do not have to be different; the purpose of Point 3 is to ensure that $\mathrm{Im}(g) \subseteq \mathrm{Im}(f)$.
> 6. No, it is the linear ordering induced by the policy evaluation function $J$. That is, $R_1$ and $R_2$ are considered to have the same ordering of policies if $J_1(\pi_1) \leq J_1(\pi_2)$ if and only if $J_2(\pi_1) \leq J_2(\pi_2)$.
> 7. Any reward function can be viewed as an $|S|^2|A|$-dimensional vector. We use $||x,y||_2$ to denote the metric induced by the $\ell_2$-norm. This means that $||R, t_n(R)||_2$ is the $\ell_2$-distance between the two reward functions $R$ and $t_n(R)$. Since several reviewers found this notation confusing, we will replace it with more typical notation.
> 8. Cumulative Prospect Theory is not covered by our analysis in this paper, but we consider it (along with other models from the behavioural sciences, such as hyperbolic discounting) to be an important topic for further work. We would however expect the worst-case error to be very large in this case, and proof strategies similar to those used in Section 3.3 could probably be used. Note that Cumulative Prospect Theory can lead to non-stationary policies.
>
> We hope that these points clarify our contributions, and that the reviewer will consider increasing their score!
>
>
> References:
>
> Joey Hong, Kush Bhatia, and Anca Dragan. On the sensitivity of reward inference to misspecified human models, 2022.
>
> Joar Skalse and Alessandro Abate. Misspecification in inverse reinforcement learning, 2023.
>
> Rachel Freedman, Rohin Shah, and Anca Dragan. Choice set misspecification in reward inference, 2020.
>
> Luca Viano, Yu-Ting Huang, Parameswaran Kamalaruban, Adrian Weller, and Volkan Cevher. Robust inverse reinforcement learning under transition dynamics mismatch, 2021.

---

> > ### Author Response · Authors · 2023-11-12
> > **Comparison to Existing Work**
> >
> > The relationship between our work and previous work in the same area can be summarised as follows:
> >
> > Freedman et al, 2020 and Viano et al, 2021 study two specific forms of misspecification in IRL, namely choice set misspecification and environment dynamics misspecification. By contrast, our analysis covers many different forms of misspecification --- our Theorem 1 and Corollary 1 provides necessary and sufficient conditions that cover all types of misspecification, and we also investigate perturbation robustness and misspecified discount functions in more depth. Neither Freedman et al, 2020 nor Viano et al, 2021 cover these forms of misspecification.
> >
> > Skalse & Abate, 2023 study many different kinds of misspecification, and like us, they also aim to derive necessary and sufficient conditions that describe all forms of permissible misspecification. However, they measure the distance between reward functions using equivalence classes. This is a serious limitation, because it means that their their analysis is unable to distinguish between large and small errors in the inferred reward. By contrast, we use metrics, which allows for a more fine-grained analysis.
> >
> > Hong et al, 2022 measure the distance between policies using the $\ell_2$ norm, assume that the behavioural model satisfies strong log-concavity, and assume that there is a unique reward function that maximises fit to the training data. Using the $\ell_2$-norm is problematic, as we briefly discuss in Secton 2.2, and the latter two conditions are often not satisfied in practice. These factors limit the extent to which their conclusions are applicable to real-world cases. By contrast, we use a metric with stronger theoretical guarantees, and we carry out our analysis in terms of the behavioural models that actually are common in contemporary IRL. Moreover, we provide necessary and sufficient conditions for misspecification robustness (in Section 3.1), which is not done by Hong et al, 2022.

---

> > > ### Comment · Reviewer_Bsty · 2023-11-22
> > >
> > > I thank the authors for the detailed response to my questions. While most of my questions have been addressed, I still feel that some numerical experiments to demonstrate the utility and impact of the analyses would be needed. I am increasing my score to reflect this.

---

### Official Review · Reviewer_5VPU · 2023-11-02

**Soundness:** 3 good
**Presentation:** 3 good
**Contribution:** 3 good
**Rating:** 8
**Confidence:** 3

**Summary:**

The authors investigate the question of robustness of an IRL algorithm against reward misspecification, i.e. if the expert follows a behavioural model (which computes $\pi$ from $r$) and an IRL algorithm assumes another behavioural model, when will the inferred reward be close to the true reward? To understand this, the authors use the STARC metric [1] to define closeness between two given rewards. The authors then identify necessary & sufficient conditions for robustness to hold when misspecification exists, and further discuss parameter misspecification, continuity of the behavioural model, etc. Overall, the paper is well written and provides useful insights fundamental to IRL, but is still less clear how these insights translate to practical rules for IRL practitioners.

**Strengths:**

Lots of good explanations, especially in the appendix

**Weaknesses:**

Understanding the results in this paper takes multiple readings, and it is difficult to follow in a few places. This is due to choices that seem arbitrary, but are well explained later. For example, the choice of $sin(2arcsin(\epsilon/2))$ in Proposition 3 seems arbitrary, but is justified in the Appendix. Overall, the writing is good, but can be improved by providing more intuition about the theorems/propositions in the main text, possibly through more examples, rather than relegating all explanations to the appendix.

**Questions:**

1. What if we have two IRL algorithms that differ in implementation but assume the same underlying behavioural model? Will this affect the results?
2. Does the IRL reward parameterization affect the results?
3. Are there more ways in which reward functions can be different but yield the same optimal policy? (except for potential shaping, S' redistribution, positive linear scaling)
4. Shouldn't a reward change of the type $R'=R+b$ where $b$ is a constant also lead to the same optimal policy? Is this discussed somewhere (may have missed it)?
5. Can these results be used to quantify unidentifiability better? For example, unidentifiabilty in RL/IRL states that there may exist multiple rewards that yield the same policy, so given a policy, an IRL algorithm can output multiple valid reward functions. But what is the size of a class of rewards that are equivalent in some way? What is this size affected by?

**Typos**
- Page 2, Paragraph 2, Line 6: "misspecifiaction" => "misspecification"
- Section 3 title: "rubustness" => "robustness"

**References**
1. Starc: A general framework for quantifying differences between reward functions, Skalse et al. (2023)

**Details Of Ethics Concerns:**

_

---

> ### Author Response · Authors · 2023-11-12
> **Response**
>
> We thank reviewer 5VPU for their review and feedback!
>
> We will try to include more examples in the main text. Due to space restrictions, it is difficult to give in-depth explanations without compromising on the technical content. However, since several reviewers have commented on Proposition 3 in particular, we will prioritise more intuition building for this result.
>
> Questions:
> 1. Our results apply to any algorithm which uses a particular behavioural model, so our results are directly applicable in this case. However, note that if we make additional assumptions about both the true reward function, *and* about the inductive bias of the learning algorithm, then we may be able to derive stronger results. This is discussed in Appendix A.4.
> 2. This might affect the results if the reward is parameterised in a way that cannot express all possible reward functions -- this is discussed in the third paragraph of Section 2.1, and in Appendix A.3. If the parameterisation can express all possible reward functions, then it will not affect the results.
> 3. Yes. Potential shaping, S'-redistribution, and positive linear scaling together completely characterise all ways in which two reward functions can have the same *policy order*. However, two reward functions can have different policy orderings, and still have the same optimal policy. To see this, consider a Bandit MDP with three actions, and two reward functions $\langle 2, 1, 0\rangle$, $\langle 2, 0, 1\rangle$. These reward functions have the same optimal policy, but different policy orderings (and so they do not differ by potential shaping, S'-redistribution, and positive linear scaling).
> 4. These types of transformations can be expressed as potential shaping, just let $\Phi(s) = -b/(1-\gamma)$ for all states $s$.
> 5. Yes, the techniques we have used can also be used to reason about unidentifiability. For example, proposition 1 and 2, together with the properties of STARC metrics, imply that if two reward functions produce the same Boltzmann-rational or MCE policy, then their STARC distance is 0. This means that an IRL algorithm that is based on the Boltzmann-rational or the MCE behavioural model will converge to a reward function that has STARC distance 0 to the true reward function (when there is no misspecification).
>
> We also thank the reviewer for notifying us of the typos!

---

> > ### Comment · Reviewer_5VPU · 2023-11-22
> > **Rebuttal response**
> >
> > Thank you for the clarifications. I am inclined to keep my score in favor of acceptance.

---

### Meta-Review · Area_Chair_dF5w · 2023-12-12

**Metareview:**

This paper explores the sensitivity of Inverse Reinforcement Learning (IRL) to misspecification of behavioral models, revealing necessary and sufficient conditions for observed data deviations without exceeding a specified error threshold. The analysis highlights the substantial sensitivity of the IRL problem to even mild misspecifications, emphasizing potential errors in inferred reward functions. As most of the reviewers agreed that paper indeed makes notable progress toward theoretical understandings of misspecification in IRL, conducts a systematic analysis and provides complete answers to this problem. Its also well written so we accept the submission.

Please incorporate all the reviewer comments in the final version of the paper.

**Justification For Why Not Higher Score:**

N/A

**Justification For Why Not Lower Score:**

N/A

---

### Decision · Program_Chairs · 2024-01-16

Accept (poster)